# In situ assembly of bioresorbable organic bioelectronics in the brain

Martin Hjort[1], Abdelrazek H. Mousa [2], David Bliman [2],
Muhammad Anwar Shameem[2], Karin Hellman [1], Amit Singh Yadav[1],
Peter Ekström [1], Fredrik Ek [1] & Roger Olsson [1,2] ✉

Bioelectronics can potentially complement classical therapies in nonchronic treatments, such as immunotherapy and cancer. In addition to functionality, minimally invasive implantation methods and bioresorbable materials are central to nonchronic treatments. The latter avoids the need for surgical removal after disease relief. Self-organizing substrate-free organic electrodes meet these criteria and integrate seamlessly into dynamic biological systems in ways difficult for classical rigid solid-state electronics. Here we place bioresorbable electrodes with a brain-matched shear modulus—made from water-dispersed nanoparticles in the brain—in the targeted area using a capillary thinner than a human hair. Thereafter, we show that an optional auxiliary module grows dendrites from the installed conductive structure to seamlessly embed neurons and modify the electrode's volume properties. We demonstrate that these soft electrodes set off a controlled cellular response in the brain when relaying external stimuli and that the biocompatible materials show no tissue damage after bioresorption. These findings encourage further investigation of temporary organic bioelectronics for nonchronic treatments assembled in vivo.

In vivo assembled, transient, and bioresorbable bioelectronics have the potential to supplement pharmaceuticals in nonchronic treatments (e.g., immunotherapy, pain, and cancer). Conventional drugs typically solely address biochemical processes, whereas bioelectronics tackle the dysfunction of bioelectric circuits. Research on conductive structures in vivo mainly focuses on chronic applications. In transient bioelectronics, minimally invasive methods for implantation and bioresorbable materials are central requirements in addition to function[1]. The latter is because there is no need for revision surgery after treatment using transient materials. This would, for example, be important in the electrotherapy of solid tumors, especially in the brain, such as glioblastoma[2,3].

Various strategies have been undertaken to design electrodes that seamlessly connect to the structures of the nervous system in animals[4–8]. Most depend on external cues, such as electric energies, chemicals (including proteins), or genetic engineering. In pioneering work by Martin and co-workers, conducting polymers were formed within rodents' brains. A solution of the monomer (3,4-ethylenedioxythiophene [EDOT]) and the larger polymer polystyrene sulfonate (PSS), acting as a template, was injected and electropolymerized in vivo to derive a massive spheric cloud protruding from the electrode[9]. To further advance the concept of in vivo polymerization, the genetic engineering of animals was carried out to make enzymes expressed in specific cells that promote the local polymerization of aniline near the target tissue in nematode *C. elegans* and mice[6]. However, from this study, it is not easy to evaluate the importance of the expression specificity of the enzymes in vivo. After the syringe injection of aniline, a dense and comparably large hemisphere of polymers was formed in the mouse brain. This is probably because of diffusion-controlled polymerization. Although elegant, the dependence on genetic engineering is not optimal for nonchronic therapies and complicates its use in humans. To avoid genetic manipulation, the

[1]Chemical Biology & Therapeutics, Department of Experimental Medical Science, Lund University, SE-221 84 Lund, Sweden. [2]Department of Chemistry and Molecular Biology, University of Gothenburg, SE-405 30 Gothenburg, Sweden. ✉e-mail: roger.olsson@med.lu.se

enzymes that catalyze polymerization could be endogenously expressed. Endogenous peroxidase enzymes polymerize 4-(2-(2,5-bis(2,3-dihydrothieno[3,4-*b*][1,4]dioxin-5-yl)thiophene-3-yl)ethoxy)butane-1-sulfonate (ETE-S, Fig. 1a) in plants[10] and the tiny freshwater organism *Hydra vulgaris*[11]. However, this promising approach is limited to tissues that exhibit sufficient peroxidase activity, and no mammalian peroxidases have been reported to form conductive polymers in vivo. In 2023, we solved this problem by in vivo injecting a cocktail including an oxidase enzyme and horseradish peroxidase (HRP), where the oxidase consumed endogenous metabolites while generating hydrogen peroxide for HRP-mediated polymerization of an ETE-S analog, ETE-COOH, into a conductive structure[8]. However, these enzymatic in vivo formed conductive polymers are used in passive mode; they are not affected by direct external stimuli. Thus, combining the formation of seamless cell-integrating organic microstructures with direct external stimuli has proven difficult.

Our exploratory research used vertebrate zebrafish (*Danio rerio*) as a model. Zebrafish have proven to be translationally relevant; they possess evolutionary conserved features, ranging from gross brain structure to behavior. Using zebrafish is cost-efficient and regarded as a 3R (reduce, refine, and replace) alternative, making ethical considerations more defendable than exploratory studies in mammals. Allometric scaling to larger animals and humans is possible in compliance with traditional drug development[12].

Recently, we reported on the discovery of A5, a self-doped water-soluble mixed ion–electron conductor poly(3,4-ethylenedioxythiophene)butoxy-1-sulfonate (PEDOT-S) derivative (Fig. 1a)[13]. Among the PEDOT-S derivatives, A5 is unique because it self-assembles in an agarose gel cast with a physiological buffer and generates a highly conductive hydrogel (1–5 S cm$^{-1}$) stable for several months. Furthermore, on average, A5 comprises small polymers (i.e., oligomers) of 7–8 monomers. Thus, A5 is smaller than the antisense oligonucleotide drugs (about 20 nucleotides) and is expected to have better bioresorption properties than PEDOT:PSS, where the PSS part is a large polymer of 200–300 monomers ($M_n$ ~70,000 g mol$^{-1}$).

Here, we report a general approach that is not dependent on specific external or endogenous triggers to assemble bioresorbable high-performing electrode structures within the central nervous system (CNS) implanted using a minimally invasive method. We show that highly water-dispersed nanoparticles—comprising the conductive polymer A5—injected into a zebrafish brain using a small-diameter capillary self-organize into a mixed ion–electron conducting hydrogel. The conductive hydrogel is transient, and the initial inflammation in the brain caused by the injection resolves, leaving no tissue damage from the electrode behind. To modify the initial conductive structure after injection, ETE-R derivatives (e.g., ETE-S and ETE-PC, Fig. 1a) were mixed into the A5 solution, injected into the brains, and electropolymerized. ETE-Rs provide the functionalization of choice and facilitate reaching neurons where the initially formed electrodes would not. The ETE-Rs are designed to have a lower oxidation potential than the EDOT monomer: 0.3–0.5 V and 1.2 V, respectively, which is significant for minimal damage to the tissue during

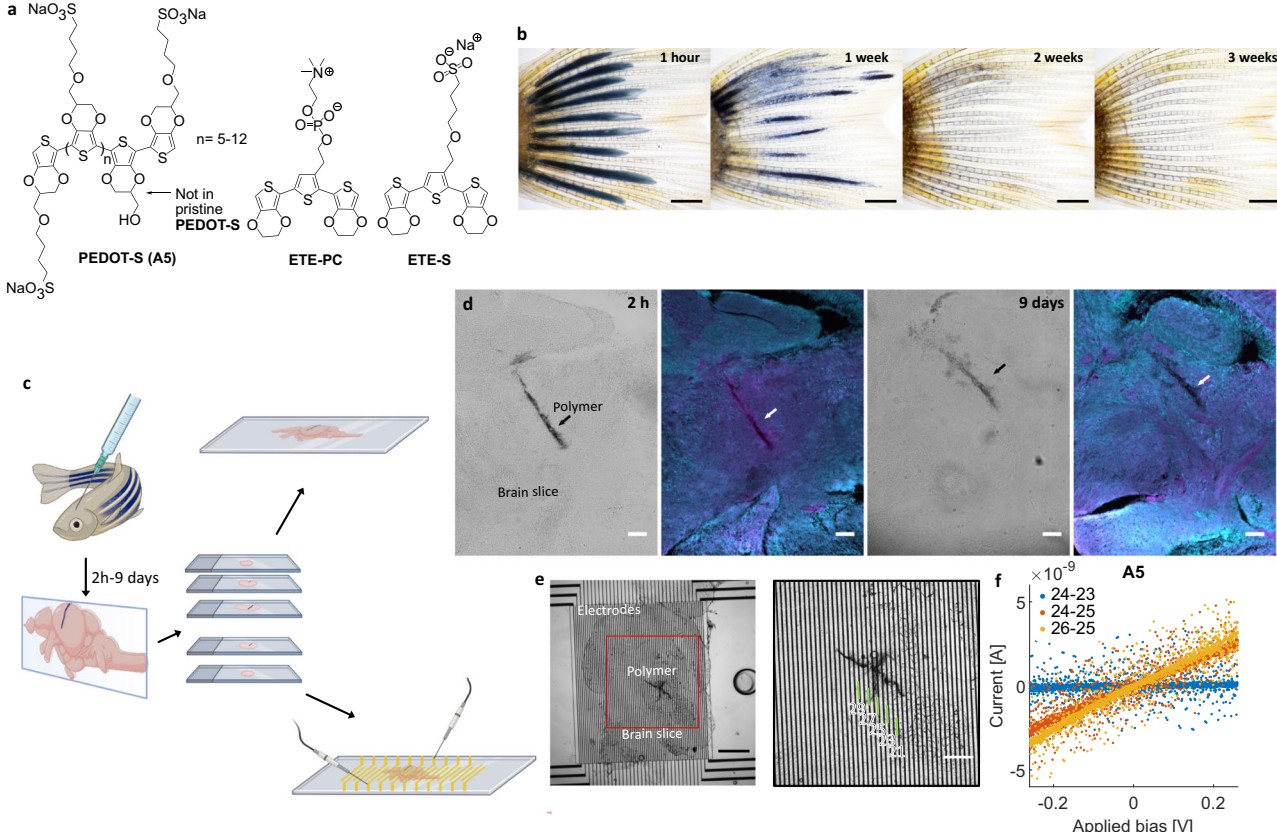

**Fig. 1 | The PEDOT-S derivative A5 was used as an in vivo injectable conductive polymer. a** Chemical structure of A5 and the two ETE-R trimers ETE-S and ETE-PC. **b** Time-lapse following the microinjection of A5 into the tailfin of an albino zebrafish. Images were obtained from the same fish after 1 h, 1 week, 2 weeks, and 3 weeks. *n* = 3 fish. **c** Workflow describing A5 microinjection into live zebrafish brains, including brain excision and cryosectioning. **d** Microscopy images depicting brain slices with part of the polymer track as observed in zebrafish that were left to swim for 2 h or 9 days after injection; bright-field image to the left with fluorescence image of the same slice showing the RedOx staining, where the magenta shows inflammation from the initial injection, but not from the polymer (2 h: *n* = 6 fish; 9 days: *n* = 9 fish). **e** Electrical measurements of brain slices placed on interdigitated Au electrodes. Au electrode numbers are indicated. *n* = 3 fish. **f** Current–voltage sweeps obtained between the contacts shown in the figure. Scale bars: 1 mm (**b**), 100 μm (**d**), 500 μm (**e**), 100 μm (**f**). Figure created with BioRender.com.

electropolymerization. In addition to the structural modification (shape and functionality) and increased in vivo stability, conductivities of two to three orders of magnitude higher than tissue were measured.

## Results

### A5 self-assembles to form a conductive electrode in tissue

Low-concentration agarose gel cast with physiological buffer (Ringer's solution) mimics brain tissue and is an excellent introduction to understanding the aggregation behaviors of self-organizing conductive polymers. Divalent ions can enable A5 to self-assemble into a long-term stable hydrogel in Ringer's solution–agarose gel showing significantly higher conductivity than the surrounding environment[13]. This inspired exploration of the use of A5 as an injectable in vivo electrode. Although tissue mimics and cell cultures are beneficial and ethically ideal for establishing basic properties, highly dynamic homeostasis and cellular diversity in vivo are challenging to study in vitro. In this exploratory study, the 3R vertebrate zebrafish model was used for in vivo studies; it is ethically sound and translationally relevant.

The zebrafish caudal fin is a model system for limb regeneration and neuropathy–it is highly dynamic. It is transparent, enabling direct optical access to an injected polymer. A5 (20 mg mL⁻¹) was injected into the caudal fin of zebrafish between the fin rays (Fig. 1b). To facilitate the assembly of the soft electrodes in the fin, a 25% Ringer's solution was used to dissolve the A5. This use of a formulation with additional ion strength, compared to Milli-Q water, is not necessary when injecting into the zebrafish brain, as will be shown below. However, this exemplifies the adaptive nature of A5, matching the nanoparticle formulation with specific tissue and making it possible to inject into lower and higher ionic strength regions. Directly after injections, dark-blue coherent structures were formed between the fin rays (Fig. 1b).

The A5 was contacted using two tungsten microelectrodes connected to a Keithley sourcemeter to measure the gel's resistance (Fig. S1, Fig. S8). This proved challenging as the A5 gel was pushed around by the W-electrodes leading to poor W–gel contact. Microampere currents were measured (resistance 0.16 MΩ) in the A5 gel; this is about twice the current compared to the reference rays without A5 (0.32 MΩ), Fig. S1. The difference was more pronounced after drying the fin, reaching a difference of two orders of magnitude. This is because of a decreased resistance of the A5 (0.02 MΩ) and an increased resistance of the reference fin rays (1 MΩ), which was not surprising since the conductive A5 oligomers were expected to come closer together upon drying, thereby allowing for higher conductivity.

The conductive structure had partly disappeared after one week for fish left to swim with A5 in their caudal fins. It was completely gone after three weeks (Fig. 1b). Throughout the experiment, no changes in zebrafish behavior or fin damage were observed. Thus, the electrodes were transient and bioresorbed, leaving no visible damage.

The brain is the most complex and delicate biological system and challenging to correctly mimic in surrogate models. Therefore, A5 was injected into the brain for direct evaluation in the targeted environment (Fig. 1c). For studies in zebrafish brains, A5 (20 mg mL⁻¹) was injected using the columnar injection technique, a method used for intracerebral cell therapy in humans. Cannula diameters of 700 μm have been used for cell therapy where injected cells are in the order of 10–20 μm in diameter. However, to avoid blood vessel rupture and bleeding in the brain, it was recently suggested that a syringe diameter below 25 μm should be used to implant bioelectronics[14]. A5 nanoparticles are highly water-dispersed and have an average diameter of 80 nm[13], making it possible to use cannulas in the suggested size range. In this study, we used a diameter of 30 μm, which was within the margin of error of the recommended size. Immediately after injection, the fish were revived and allowed to swim for up to nine days, showing no detrimental effects from the injected polymer.

A redox histochemistry method was used to evaluate the response to the oxidative insults of injections and the soft electrode formed on brain tissue, a technique that allows for sensitive detection of changes in the intracellular redox state in situ; thiol groups in glutathione and proteins sense the cellular redox environment[15]. Two hours post-injection, an extensive inflammatory response was observed around the injection area (Fig. 1d). The inflammation was resolved nine days after injection, with A5 still present, thus indicating that the inflammation was mainly induced by tissue damage from the columnar injection. At the latter point, the conductive structure started to degrade and lacked long-distance conductivity. However, no change in the redox state compared to normal tissue was observed in the degradation and bioresorption process.

After establishing the biocompatibility, geometry, and bioresorbable properties of the injected polymer, the conductive properties within the brain tissue were investigated. Fish with polymer were left to swim for up to seven days, after which the brains were excised, sectioned, and placed on interdigitated gold electrodes (Fig. 1e). In consecutive order, two interdigitated electrodes were biased (Keithley) and the current was registered. Although small, defined patterns of conductive polymer were visible, about 50 μm track widths and several hundred μm in length could easily be seen in bright-field microscopy as dark regions. Enabling the detection of low currents reliably.

A5 has previously shown an excellent in vitro conductivity of more than 30 S cm⁻¹[13]. However, the conductivity is expected to be much lower in complex biological systems, such as the brain, where the conductive polymer can diffuse, be sequestered by endogenous mechanisms, removed by active transport, intercalated with endogenous chemical species, or combinations thereof. Further technical challenges added additional complexity, such as variabilities in Au electrode–tissue section contact resistance and poorly defined geometries. Still, we could clearly observe a linear current–voltage dependence between close-lying contacts (15 μm) on the brain slices, thus indicative of a highly conductive polymer. A5 presented currents in the range of a few nA under biases of up to 0.2 V (resistances of 50–100 MΩ). For brain regions without A5 or non-continuous A5, see the blue trace in Fig. 1f; as expected, we could not measure any electrical conductivity. We typically could not measure longer distance conductivities, making it difficult to extract a value for the conductivity. However, our resistance values compare favorably to a recently published study on genetically targeted in vivo conductive polymer assembly where Liu et al. used up to 100 V (we used 0.2 V) to measure polymers with a resistance of more than 10 GΩ[6].

### Electrofunctionalizing A5

Using the columnar injection technology gives essential control over the positioning and pattern of the soft electrode in the tissue, perhaps more so than the assembly controlled by the genetic expression of enzymes. To add modularity to the A5 electrode–tissue interface, an auxiliary method that enables flexibility in the electrode surface area, functionality, and further protrusion for seamless extension into the tissue was developed. Co-injecting the aforementioned ETE-R derivatives (e.g., ETE-S) with A5 would position the soft electrode at the targeted site. Because of the concentration gradient and the electrostatic repulsion between the negatively charged ETE-S and the negatively charged A5, the former diffused from the formed A5 electrode. After a delay, electropolymerization using A5 as an electrode attached the ETE-S to the surface of A5 and intercalated within the A5 backbone.

The ETE-R derivatives have significantly lower oxidation potentials than EDOT, -0.3–0.5 V and 1.2 V vs. Ag/AgCl, respectively; the latter is close to the oxidation potential of water[16]. Thus, the electropolymerization of ETE-R derivatives is mild and not potentially toxic to tissues. In addition to increasing the surface area and tissue reach, different substituents (R) on the ETE-R derivatives would change the properties of the A5 electrode, as exemplified herein by ETE-S and the

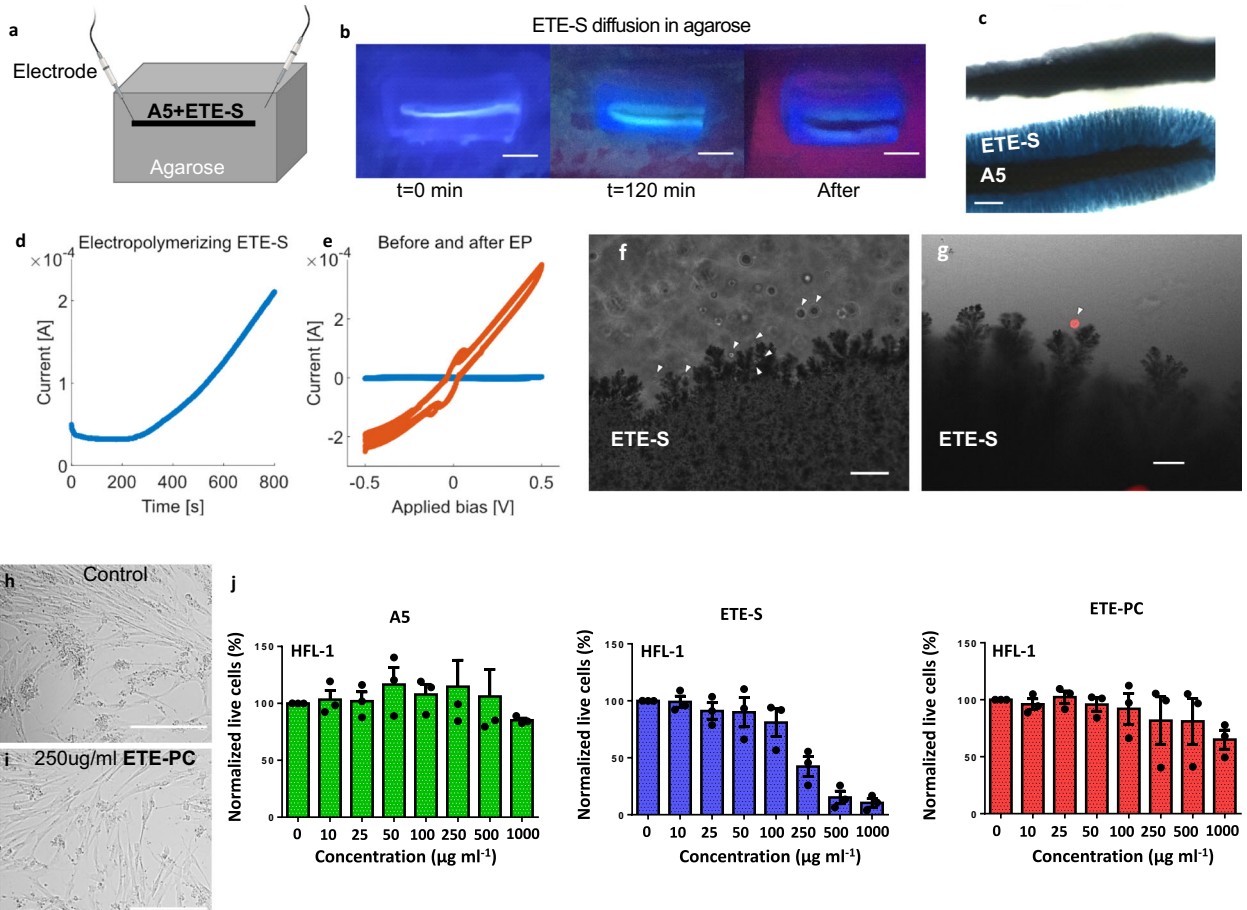

**Fig. 2 | Electrofunctionalizing of A5 with ETE-S in vitro. a** Schematic depicting the in vitro electropolymerization of the A5 + ETE-S solution after injection into an agarose mold. **b** Monitoring ETE-S diffusion using UV-light (365 nm). The experiment was repeated >5 times. **c** A5 in agarose before and after electropolymerization with ETE-S. The experiment was repeated >5 times. **d** Time-dependent current during A5–ETE-S electropolymerization with both contacts on the A5 core. **e** Smoothed Current–Voltage sweeps before (blue) and after (orange) A5–ETE-S electropolymerization in agarose with both contacts on the A5 core. **f**, **g** Electropolymerized A5–ETE-S in agarose containing A549 cells with (**g**) or without (**f**) DiI cell labeling (red channel). The positions of some cells are indicated by white arrows. The experiment was repeated 3 times. **h**, **i** Bright-field microscopy images depicting HLF−1 cells exposed to ETE-PC at 0 μg ml⁻¹ and 250 μg ml⁻¹, respectively. **j** Normalized live HLF-1 cell count upon exposure to A5, ETE-S, or ETE-PC at varying concentrations (n = 3 technical and 3 biological replicates). Data are presented as mean values ± SEM. Scale bars: 5 mm (**b**), 1 mm (**c**), 100 μm (**f**), 50 μm (**e**), 500 μm (**h**, **i**). Image created with BioRender.com.

zwitterionic ETE-phosphatidylcholine (ETE-PC, Fig. 1a). The required high solubility of the injectable electrode imposes requirements on the nanoparticles forming the soft electrode, namely that they should be highly water-soluble and still be able to self-organize when injected into the tissue and thereafter bioresorbable without harming tissue. Thus, instead of redesigning the oligomers forming A5, this modular approach takes advantage of unique A5 properties and then adds soluble monomers in situ to customize the electrode–tissue interface.

The auxiliary A5 modular concept was evaluated in an in vitro agarose gel (0.5%) cast with Ringer's solution, which mimics brain tissue. A5 (20 mg mL⁻¹) was dissolved in ETE-S or ETE-PC (40 mg mL⁻¹), without forming any precipitate. The dark solution was injected into agarose using a Hamilton syringe (Fig. 2a) where the A5 material instantly formed an aggregate. The diffusion of the ETE-R derivatives from the A5 injection track was monitored using UV light (365 nm, Fig. 2b). After 2 h, ETE-R diffused from the A5 at a distance approximately twice the thickness of the A5 hydrogel electrode (Fig. S9). The A5 was then the connected electrode, and the ETE-Rs were electro-functionalized at an applied bias of 1.5 V vs Au counter electrode (0.5 V vs Ag/AgCl counter electrode). This enabled electropolymerization, even with possible contact resistance. The applied bias can be further optimized, but it was selected for further use to have a high tolerance to variances in contact resistances during electropolymerization.

The thickness of the A5 electrode increased, confirming the successful polymerization of ETE-S (Fig. 2b). Further image analysis showed dendritic structures growing from the A5 core (Fig. 2c and Movie S6). The electropolymerized regions also displayed increased conductivity (Fig. 2d, e). With one contact on the A5 (and the other in the agarose) during electropolymerization, the comparably high resistance in agarose ensures a limited, constant current flow, regardless of electrode spacing (Fig. S11). Instead, by contacting both electrodes on the A5 during electropolymerization, the decreased resistance can be directly mapped as an increase in the current flow (Fig. 2d). The current increased by an order of magnitude during polymerization over a time course of 12 min. No change in the geometry, except for dendrite formation, of the A5 core was observed; the decreased resistance was explained by the polymerized ETE-S intercalating between the A5 nanoparticles. Although we optically observed electropolymerization (A5 darkening) when applying the voltage, we did not register a corresponding increase in current flow for the first 5 min. The initial incubation time could arise from one or more resistive bottlenecks that limit the current. Once these bottlenecks were removed by electropolymerization, a gradual current increase was observed as electropolymerization continued uniformly along the A5. Keeping both ends of the A5 contacted allowed for cyclic voltametric measurements, which showed that electropolymerization increased the current 100 times (Fig. 2e).

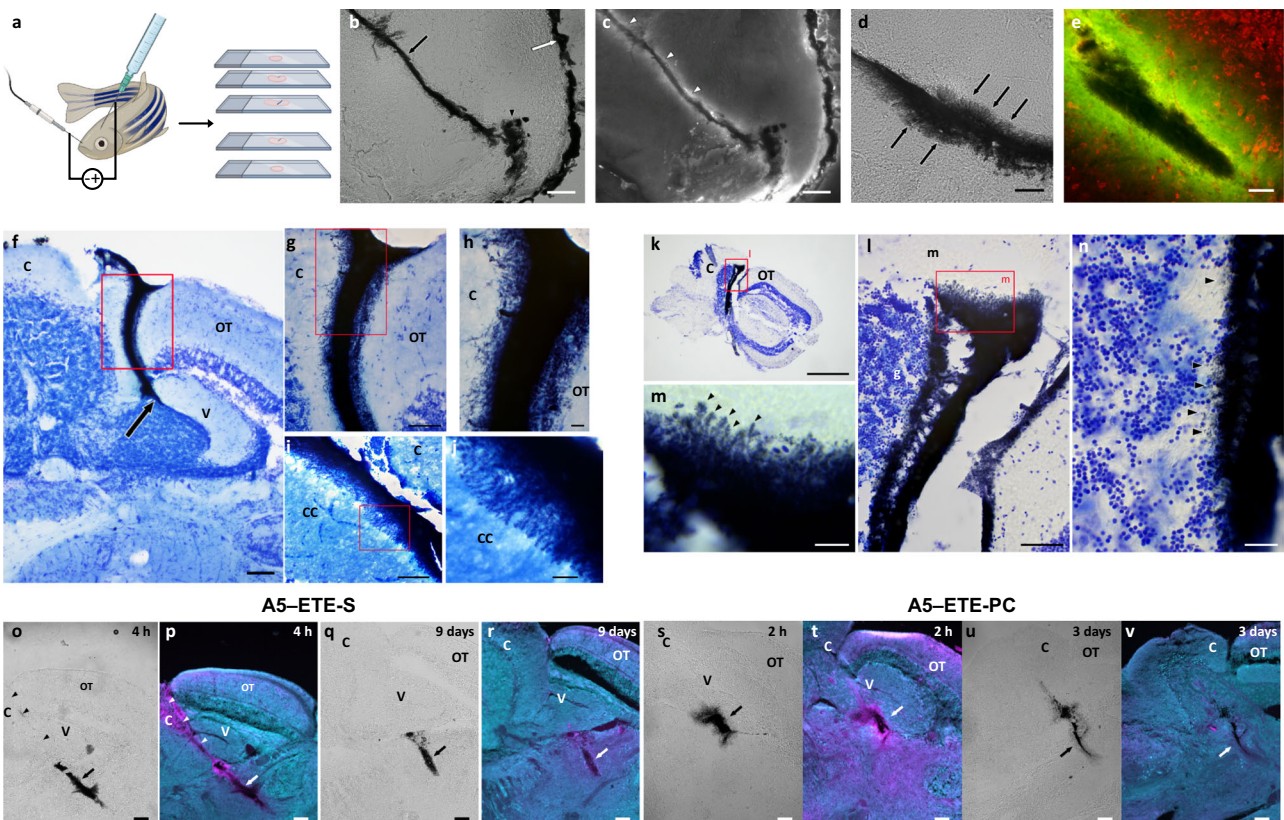

**Fig. 3 | In vivo A5 electrofunctionalizing using ETE-R in zebrafish brains.**
**a** Schematic showing the procedure. **b, c** A5–ETE-S with short electropolymeriza-
tion duration (1 min). A5–ETE-S core (arrow) follows the injection track; ETE-S
monomer was visible as a fluorescent halo (white arrowheads in **c**). n = 6 fish.
**d, e** A5–ETE-S after longer electropolymerization (5 min), where ETE-S forms den-
drites (arrows). **e** A5–ETE-S (black) extending into surrounding brain tissue with
ETE-S monomer (green) and blood vessels with blood cells and neuronal cell bodies
(red) stained with Nissl stain NeuroTrace 640/660. n = 3 fish. **f–n** Histological
staining of A5–ETE-S and zebrafish brain using cresyl violet (**f–h**, **k**, **l–n**) or
methylene blue–thionine (**i, j**). **f** A5–ETE-S (black) injected into V, arrow marks
injection site, and spreads between the *OT* and *C*. **g, h** Higher magnification of the
area boxed in (**f**). **i** Injection between the *C* and *VL*. **j** Magnification of the area boxed
in (**i**). **k** Injection into the rostral part of *C*. **l** Higher magnification of the boxed area

in (**k**). **m** Higher magnification of the boxed area in (**l**) ETE-S dendrites (arrowheads)
extending into the molecular layer. **n** A5–ETE-S (arrowheads) extending into the
granular layer in a cryosection adjacent to the one shown in (**k–m**). **f, g**: n = 3 fish;
**i, j**: n = 3 fish; **k–n**: n = 3 fish. **o–v** RedOx staining of A5–ETE-R in zebrafish brains.
The fish were sacrificed, and the brains were immediately processed for redox
staining 4 h (**o, p**), 8 days (**q, r**), 2 h (**s, t**), or 9 days (**u, v**) after injection.
**o, q, s, u** A5–ETE-R (*arrows*) in the brain tissue is seen. Note the distinct track from
the injection capillary (arrowheads) in (**o**). **p, r, t, v** RedOx staining of the sections in
(**o, q, s, u**), where the magenta shows inflammation. A5–ETE-R is shown by white
arrows. C corpus cerebelli; CC crista cerebellaris; OT optic tectum; V valvula cer-
ebelli. Scale bars: 100 μm (**b, c**), 50 μm (**d, e**), 500 μm (**f, k**), 50 μm (**g, i, l**), 20 μm
(**h, j, n**), 10 μm (**m**), and 100 μm (**o–v**). Image created with BioRender.com. **o, p**:
n = 2 fish; **q, r**: n = 6 fish; **s, t**: n = 4 fish; **u, v**: n = 5 fish.

A5–ETE-Rs electropolymerized in agarose was used as a basis for
evaluating the mechanical properties (Fig. S7). The brain is very soft
with shear modulus of about 0.5–1kPa[17], which has been challenging to
match for conventional inorganic electrodes. Electropolymerized
A5–ETE-PC closely mimicked brain tissue with a static shear modulus
of 0.57 ± 0.1kPa, see supplemental information for details.

The A5–ETE-S was electropolymerized in an agarose gel incorpo-
rating live cells to evaluate biocompatibility. We molded lung adeno-
carcinoma cells, the A549 cell line, into an agarose gel without (Fig. 2f )
and with (Fig. 2g) contrast-enhancing DiI cell labeling. After injection
of the A5–ETE-S solution, one end of the polymer electrode was
contacted, and the grounded counter electrode was kept in agarose
(outside the A5). During electropolymerization, the newly formed den-
dritic structures of ETE-S extended from the A5 and achieved cell con-
tact and, in some cases, embedded the cells, creating close contact with
the cells without any observable detrimental effects such as loss of cell
integrity (Fig. 2f, g). A close connection between the electrode and cells
has been shown to be necessary for efficient and precise low-voltage
electrical stimulation–recording[18]. A5 and ETE-R toxicity was also eval-
uated in a limiting-dilution assay (Fig. 2h–j). Neither A5 nor ETE-PC
showed cell toxicity after 1 day of exposure to up to 1 mg mL⁻¹ A5 or ETE-
PC. On the other hand, ETE-S showed some toxicity at high

concentrations. The toxicity study covered a wide range reaching up to
more than 1000x more ETE-R than we injected during in vivo experi-
ments (around 200 μg vs. 400 ng), and a high dilution occurred when
injecting into the tissue. For the quantities used in the in vivo setting, no
cell toxicity from any of the compounds was observed. We summarize
the in vitro experiments concluding that A5-mediated electro-
functionalization enabled flexible surface modification, close contact
with cells, and significantly decreased electrical resistance.

## Electrofunctionalization in vivo

Transferring the approach to an in vivo setting (Fig. 3a) puts severe
constraints on the experimental settings: (1) small diameter injection
capillary to avoid blood vessel rupture; (2) A5, ETE-R, and A5–ETE-R all
need to be highly soluble and biocompatible; (3) applied voltages and
currents used for the electropolymerization need to be kept low to not
damage the brain tissue; and (4) the procedure needs to be quick to
avoid anesthesia related damage. The 30 μm diameter capillaries, pre-
coated with 50 nm iridium, were used to inject an A5–ETE-S solution
into the brain of anesthetized zebrafish (Fig. 3, S2a). After injection, the
ETE-S was left to diffuse into the tissue for 1 min. The coated capillary
was then used as the biased electrode to establish seamless contact
with the injected A5 (Fig. S2a). Placing the counter electrode onto the

zebrafish skin at the nostril allowed electropolymerization in the sedated fish with a low current (1–7 µA, Fig. S5), mimicking the agarose setup. The procedure, injection, and electropolymerization lasted about 10 min and after an additional 5–10 min, the fish were awake and displayed normal behavior. Typically, no fast darting or odd swim patterns were observed, nor were any difficulties with buoyancy and balance indicative of injury to the brain or stress behavior due to discomfort (e.g., pain) observed. We occasionally observed stress behaviors such as rolling or upside-down swimming in fish that required longer presurgery anesthesia than normal. However, this behavior was not linked to the injection per se. The lack of adverse events shows that the minimally invasive approach was sound and well-tolerated by the fish.

When excising the brains, it was evident that the electropolymerization rendered a darker polymer from the addition of ETE-S. In sections from brains injected with A5 and ETE-S with a short electropolymerization duration (1 min), the dark polymer was primarily contained within a dense and discrete injection track surrounded by a halo of ETE-S monomer (Fig. 3b, c). When injection was followed by longer electropolymerization (5 min), the polymer contained a darker middle region, the A5, with polymerized ETE-S dendrites extending outwards in a radial fashion (Fig. 3d, e). The similarities to electropolymerization in agarose further corroborated the generality of the method and comparable results were also observed for ETE-PC (Fig. S3). The nonpolymerized ETE-S (seen by its green fluorescence) followed the injection track and extended outwards by about 50 µm without visibly affecting tissue integrity, as evident from nearby blood vessels with blood cells and neuronal cell bodies (Fig. 3e).

Histological staining of the A5–ETE-S containing sagittal brain sections (30 µm thick) showed close contact between cells and the polymer at the polymer–cell interface. Figure 3f–h depicts an A5–ETE-S injection (dark blue) extending deep into the brain between corpus cerebelli (C) and optic tectum (OT) with ETE-S dendrites extending radially from the A5 reaching into the granular and molecular layers of C and superficial layer of OT. As in the in vitro cell–agarose model, the A5–ETE-S wrapped around the neurons, and some were even entirely surrounded by the conductive polymer electrode without any visible damage to the cells. This highlights the benefits of a gel-like microstructural electrode that allows for the exchange of metabolites and ions through the electrode to sustain cell homeostasis.

In the methylene blue–thionine staining (Fig. 3i, j), which also stains A5–ETE-S, seamless integration with no signs of tissue damage was observed of the A5–ETE-S being positioned between OT and C where the dendrites extend into the vagal lobes (VL). In Fig. 3k–n, A5–ETE-S extends dorsally into both the molecular and granular layer of C, further highlighting the possibility of reaching any subregions of the brain.

After the A5 injections, an inflammatory response (oxidation of tissue) from the tissue damage mainly induced by the mechanical injection was observed, which was resolved after a few days (Fig. 1d). Introducing the auxiliary module, it was important to elucidate whether adding ETE-Rs to A5 followed by electropolymerization would cause damage to the tissue. As with the A5 injection, significant inflammatory signatures were found around the injection track within a few hours of the injection of both A5–ETE-S and A5–ETE-PC (Fig. 3o–v). Zebrafish that were left to swim with the polymers for days showed a significant decrease, with ETE-S after 7-9 days and complete removal with ETE-PC after 3 days, of inflammation. Thus, the applied voltage used in electropolymerization did not cause further tissue oxidation. Maybe not surprisingly, the A5–ETE-S had basically the same oxidation profile as A5 itself; they share the negatively charged sulfonate functionality. The zwitterionic phosphatidylcholine A5–ETE-PC showed a slightly better profile, and the oxidation was completely resolved after 3 days. Immunohistological staining showed no signs of gliosis (Figs. S10, S12). This corroborates that the phosphatidylcholine,

as a constituent of cell membranes, has improved in vivo biocompatibility[19]. These data, that the electropolymerization of monomers (ETE-Rs) with low oxidation potential did not cause additional damage to the tissue combined with the normal behavior observed in the fish, strengthens the methodology as being minimally invasive and a suitable strategy to install soft conductive dendritic electrodes in brains.

## A5–ETE-R electrical properties in zebrafish brains

The conductivity of A5 was improved by ETE-R functionalization (Fig. 4). For fish that underwent electropolymerization, more than 10 times higher currents (resistances of 5–10 MΩ) were observed under the same applied bias (for both ETE-S and ETE-PC). It was also possible to map conductivity over long electrode distances, allowing for the deduction of a value on the conductivity. Adjusting to the increased diameter of the conductive polymer upon electropolymerization, we estimated that the conductivity was around 3 S cm$^{-1}$ for both A5–ETE-S and A5–ETE-PC. This is 2 to 3 orders higher than most tissues ($<10^{-2}$ S cm$^{-1}$).

Some fish were also left to swim around with the conductive polymer in the brain for 7 days, showing normal fish behavior (Fig. 4c, d). For A5–ETE-S and A5–ETE-PC, distinct polymers were clearly observed in the brains of two out of three fish but with lower conductivities than in the one-day experiments. Despite the clear polymer presence, only low currents with no clear voltage dependence were measured for the A5–ETE-S. This is due to the active bioresorption of some oligomers, thereby introducing high-resistance bottlenecks in the conductive structure. The A5–ETE-PC showed a linear voltage-dependent current in the low nA regime (resistance around 1 GΩ), thereby presenting a polymer still exhibiting long-distance conductivity. Interestingly, the higher 7-day conductivity of A5–ETE-PC indicates that this modification to the polymer rendered it more stable.

## A5-mediated brain stimulation

After establishing the possibility of installing and modifying bioresorbable electrodes in vivo, attention turned to using them to actively modulate the electrical activity in live brains. After in vivo injecting the mixture of A5 and ETE-S, followed by electropolymerization, the fish were left to swim for one day and then sacrificed. Brains were excised and vibratome sectioned, resulting in 300–400 µm brain slices containing the A5–ETE-S electrodes (Fig. 5). The procedure kept the brain slices viable with functional neuronal signaling. For these experiments, we used adult zebrafish Casper mutants (Tg(elav3:GCaMP6f)) to visualize, by GFP fluorescence, increases in intracellular Ca$^{2+}$ in neurons[20]. Action potential firing was seen through increased green fluorescence, which is a convenient and non-invasive way to track neural activity.

We used a thin tungsten microelectrode connected to a Keithley sourcemeter to contact and stimulate the A5–ETE-S in the brain slice (Fig. 5b). Train sequences of square voltage pulses were applied to modulate the electric activity in the brain slice with settings like those used previously[21,22]. Within a second after turning on the electrical stimulation, a response was observed in which regions close to the A5–ETE-S lit up (Fig. 5c, Movies S2 and S3). The external voltage pulses were applied only for 5 s to initiate a signaling cascade. The signaling cascade lasted up to 20 s and then reverted to the initial (prestimulation) situation (Fig. 5f). The slices were then left to recover for a couple of minutes before new pulse trains were applied to initiate new signaling (Fig. 5g–j). As the brain slices were used for stimulation for more than 1 h, we noticed a general decrease both in fluorescence intensity and in the extension of signaling. In addition, as the slices were re-stimulated, we saw a slower onset, however still starting within the first few seconds of stimulation.

Interestingly, a response from the tissue was observed in several regions, all located around the polymer and extending far away from the microelectrode that was in contact with the polymer. Reference

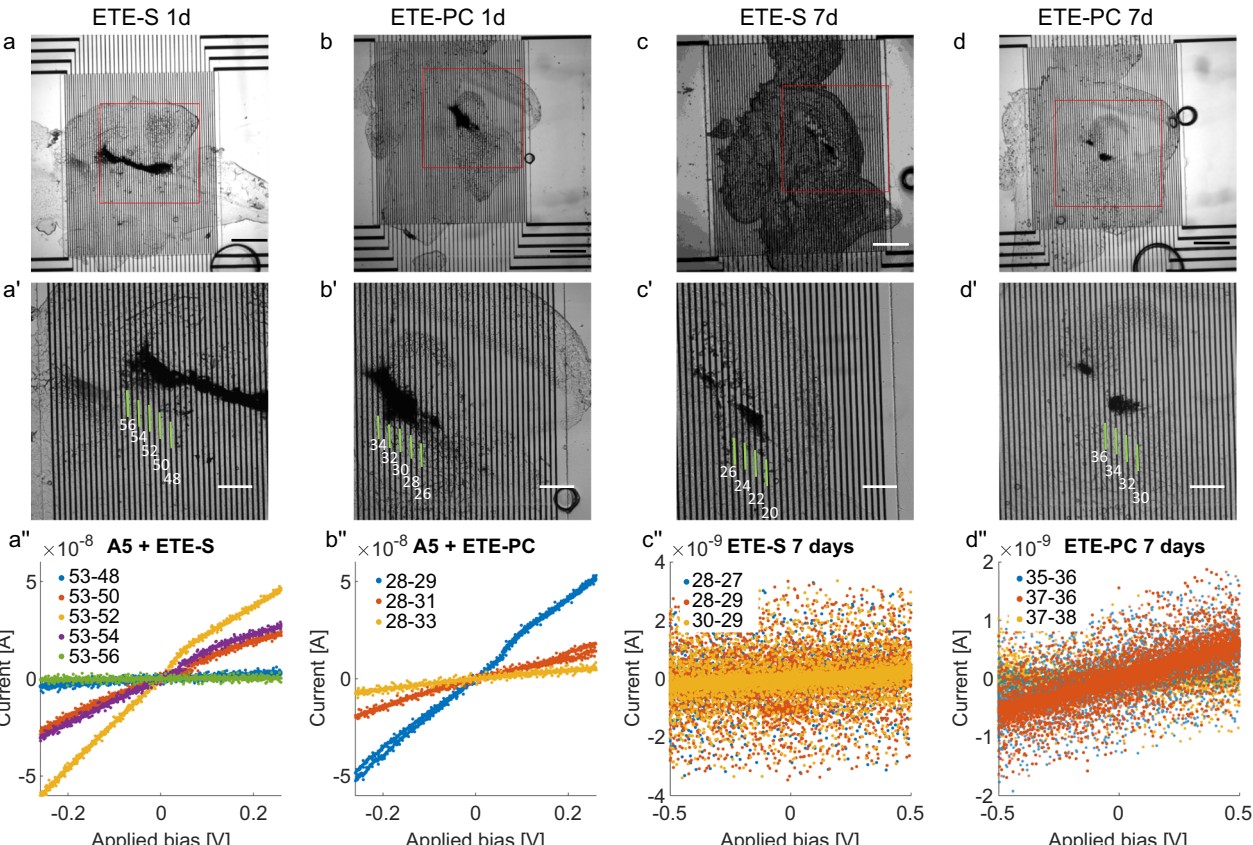

**Fig. 4 | Electrical measurements of sectioned zebrafish brains with A5–ETE-R.** Zebrafish with electropolymerized A5–ETE-R were left to swim for 1 day or 7 days, brains were excised and sectioned, and then placed on interdigitated Au electrodes for electrical measurements. **a–d, a′–d′** Microscopy images depicting brain slices on Au electrodes. Scale bars 500 μm (**a–d**) and 200 μm (**a′–d′**). Electrode numbers were overlaid in the images for clarity. **a″–d″** Electrical measurements show currents flowing between the indicated electrode leads. *n* = 3 fish in each group.

brains without A5–ETE-S did not show similar long-distance stimulation effects, showing that the conductive polymer transmits the electrical pulses to distant regions. After several rounds of stimulation, an increased spontaneous electrical activity was typically observed within the brain slices. The spontaneous activity most often appeared in the previously stimulated regions but had a different appearance in smaller, less intense firing with a stochastic appearance across the slice (Movie S3). In comparison, the externally induced excitations were more pronounced and synchronized with the onset of the applied voltage pulses.

The excited regions, mainly centered around the cerebellum and torus longitudinalis, were separated by regions without any increased neural activity, indicating the possibility of targeting specific regions in the brain. This was confirmed by adding the GABA-A receptor antagonist pentylenetetrazole (PTZ) to the imaging medium (Fig. S4). PTZ stimulates unspecific neural activity and upon addition, many more regions in the brain slices initiated neuronal firing. These regions were not close enough to the soft electrodes to be stimulated by the applied electric fields. The repeated cycling shows that stimulation was not acutely toxic to the cells, and the regional specificity highlights that the method can be implemented in an in vivo setting.

Leaving part of the microcapillary inside the brain after injection allowed to establish external contact with the soft electrode (Fig. 5k–m, S3, Movie S4). The fish were left to swim around with the capillary for one day (Movie S5) before the brains were excised, sectioned with the capillary remaining in the slices, and analyzed in the electrical setup. The stimulating electrode was positioned on the capillary outside of the brain rather than directly on the polymer in the slice (Fig. 5o). This configuration offers a solution to how one could

contact the polymer electrodes in the brains of live fish. When adding the stimulating pulse trains, the same regions became excited as when the A5–ETE-S itself was contacted. This experiment proves that not only can we use the conductive polymer to alter brain activity, but it is also possible to contact it outside of the brain, which is a critical need for most clinical applications. Performing the same workflow in freshly excised mouse brains (Fig. S6) demonstrated that the methodologies could be directly transferred to larger animals without any alteration to the protocol.

## Discussion

Herein, we have presented a technology that establishes functional and well-tolerated organic bioelectronics in the brain. To meet the demand for transient bioelectronic therapies, implantation was performed using a minimally invasive injection technique, and the formed structures were bioresorbable. The latter is a desired property in, for example, electrotherapeutic cancer treatments, making revision surgery obsolete. This was made possible by the discovery of thiophene oligomers (A5) that form nanoparticles. These nanoparticles are highly water-dispersed, which makes it possible to have them in high concentrations in a solution without aggregation. However, by injecting them into the tissue and following interaction with endogenous ions, a stable soft electrode is formed. Thus, no specific triggers are necessary. It was also possible to ion-match the nanoparticle solution with endogenous ion strengths in quite different tissues, as shown by establishing conductive structures in the zebrafish caudal fin and brain. This makes the technology general to several tissues and cross-species. Also, because the nanoparticles comprise oligomers that are the size of conventional drugs; they are bioresorbable.

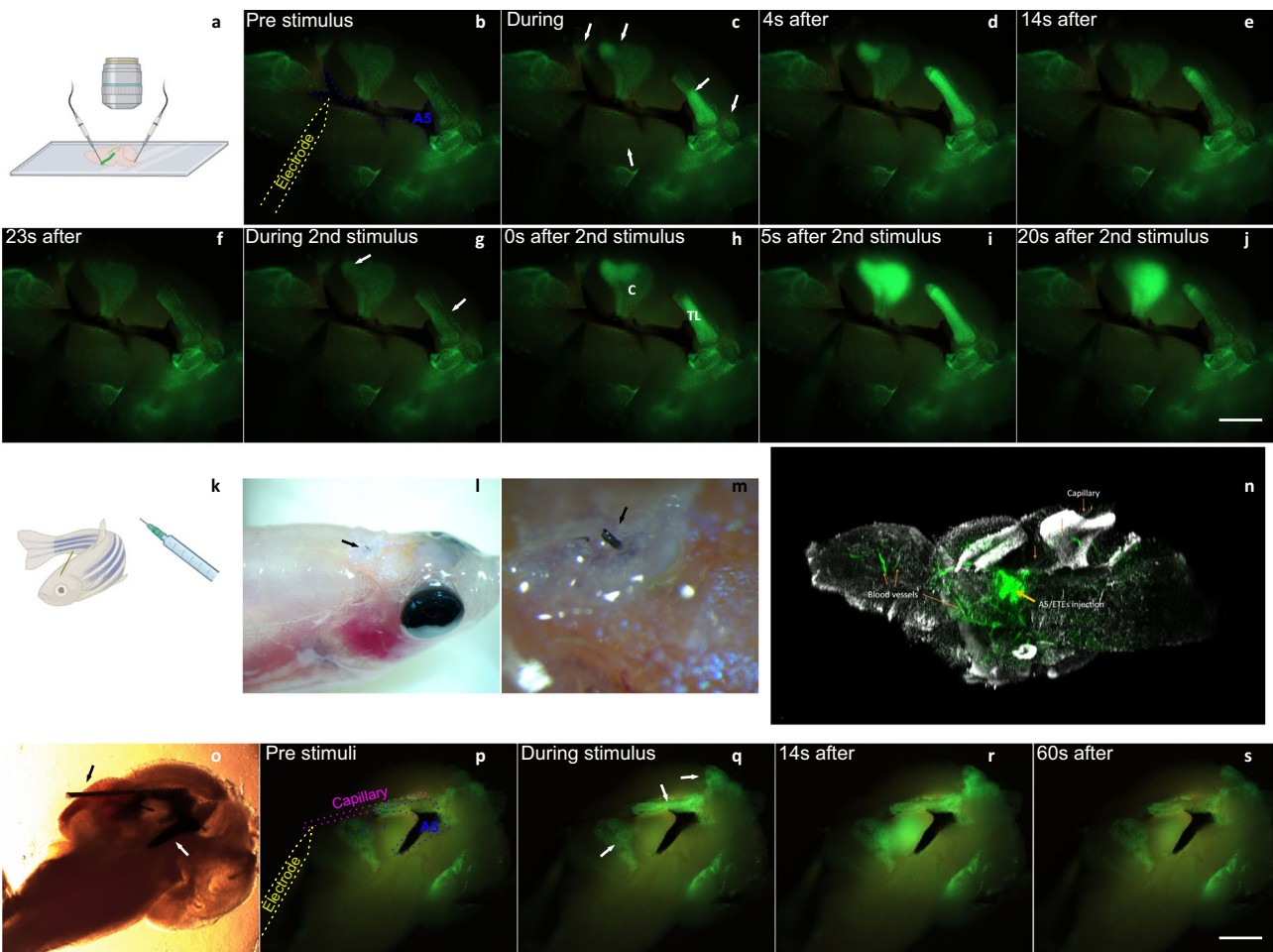

**Fig. 5 | A5-mediated electrical stimulation in live brain slices. a** Zebrafish brains with A5−ETE-S were contacted with tungsten microprobes for electrical stimulation. **b** Ca²⁺ imaging in a live brain slice with A5−ETE-S (dotted blue trace) contacted by the W-microprobe (yellow dotted trace). **c−e** When electrical stimuli were applied, several regions around the A5−ETE-S lit up (white arrows), initiating signaling cascades. **f** 23 s after excitation, the brain activity recovered, and upon a second stimulation, the brain slice was excited again (**g−j**). **k** A broken-off capillary still inside the fish brain. **l, m** Photographs of a zebrafish with capillary (black arrow) still inside the skull. **n** 3D imaging of a section with capillary still inside using light sheet microscopy showing blood vessels and ETE-S monomer (green), and neurons (white). See Movie S7. **o** Microscopy image showing a brain slice with capillary still inside the brain (black arrow) contacting A5−ETE-S (white arrow), which extends far from the capillary. **p** Ca²⁺ imaging in a live brain slice with the W microprobe (yellow) contacting the injection microcapillary (magenta) outside the brain. **q−s** When electrical stimuli were applied, several regions around the A5−ETE-S lit up (white arrows), initiating signaling cascades. In (**h**) *C* marks corpus cerebelli, *TL* marks torus longitudinalis. Scale bars 500 μm. *n* = 4 fish without capillary, *n* = 3 fish with capillary. Image created with BioRender.com.

Instead of redesigning the oligomers forming A5 to generate functional flexibility at the electrode−tissue interface, the difficult-to-copy unique properties of A5 inspired us to develop an auxiliary modular methodology. This modular approach takes advantage of A5's properties and then adds soluble monomers (ETE-Rs) that attach to A5 in situ to customize the electrode−tissue interface. The electropolymerization of ETE-Rs with low oxidation potential in situ increased conductivity, formed a close connection to cells and established functional group modifications to the A5 electrodes. This was demonstrated using both ETE-S and ETE-PC, where the latter had higher long-term stability and less toxicity. Furthermore, we showed that electropolymerization did not cause additional oxidation damage to brain tissue. This modular approach opens up for ETE-Rs with different functional groups and potentially other monomers with low oxidation potential.

Despite the small zebrafish brain, we solved the critical challenge of contacting soft neural electrodes to allow efficient external interaction. With external contact, neuronal signaling was modulated in live brain slices−excised from fish with implanted bioelectronics−by applying electrical pulses.

The methodologies and workflows presented here have been mainly evaluated using zebrafish and in mouse brain ex vivo (Fig. S6). However, a reasonable speculation is that the procedures would be more straightforward with larger brains, e.g., rodents and primates, especially external connections.

In summary, these findings initiate the discovery of in vivo assembled, fully integrated, bioresorbable electronics within nervous systems and other tissues that can be used for nonchronic treatments.

## Methods

### Animal ethics

This study was conducted in accordance with the national legislation of Sweden and with European Community guidelines for animal studies. All procedures were approved by the ethical committee in Malmö−Lund (5.8.18-05993/2018 and 5.8.18-05748/2022 adult zebrafish).

### Ringer solution

116 mM NaCl (Sigma-Aldrich M7506), 2.9 mM KCl (Sigma-Aldrich P5405), 1.8 mM CaCl₂ (Sigma-Aldrich P5405), 5 mM HEPES (Sigma-

Aldrich H3375), adjusted to pH 7.2 using NaOH (1 M) (Sigma-Aldrich S5881) or HCl (1 M) (Applichem A0659).

### E3 stock (1 L 10x)

2.9 g NaCl (Sigma-Aldrich M7506), 0.49 g $CaCl_2$ (Sigma-Aldrich P5405), 0.13 g KCl (Sigma-Aldrich P5405), 0.81 g $MgSO_4$ (Sigma-Aldrich M7506).

### E3 medium (1 L)

500 mL MilliQwater, 100 mL E3 10x, 400 mL system water from the fish facility, 100 µl Methylene blue (Sigma-Aldrich 319112).

### Tricaine stock solution (4 mg/mL)

400 mg tricaine powder (Ethyl 3-aminobenzoate methanesulfonate [Sigma-Aldrich E10521]) was added to 97.9 ml doubled-distilled water and 2.1 ml 1 M Trizma Bas (pH 9) (Sigma-Aldrich T1503). The pH was adjusted to 7 using HCl (1 M). The solution is stored in the freezer.

### Evaluation of the A5–ETE-S agarose mold

A solution of A5 (20 mg mL$^{-1}$) and ETE-S (40 mg mL$^{-1}$), synthesized according to Mousa et al.[13] and Gerasimov et al.[16], respectively, in $H_2O$ was injected into an agarose mold (0.5% agarose [Agarose, LE, Analytical Grade, Promega Corporation] in Ringer solution). Diffusion of the ETE-S was monitored using a UV lamp at 365 nm. After 2 h, one of the Au-coated W-electrodes was connected to the A5 aggregate (and one in the agarose mold to electropolymerize ETE-S to get 100% coverage (Fig. 2). ETE-S was electropolymerized using 1.5 V vs. Au counter electrode (Keithley sourcemeter 2612B, Keithley Instruments). The agarose was imaged using bright-field microscopy (10x and 40x objective).

The conductivity of the A5 and A5–ETE-S in agarose was measured using a two-terminal setup in which 25 µm Au-coated tungsten microprobes (Signatone, Gilroy, CA) were connected to the polymer embedded in the agarose. By sweeping an applied electric potential and registering the resulting current over different distances, conductivity can be estimated using the transmission line model.

### Evaluation of cell–polymer interaction

The experiment was performed according to the procedure above, except that A549 cells with or without a fluorescent label (CM-DiI, Thermo Fisher Scientific C7000) were dispersed in agarose before molding.

### Cell culturing

Human lung fibroblast (HLF-1) cells were maintained in DMEM (Thermo Fisher, Gibco, Cat no. 11995073) supplemented with 10% FBS (Thermo Fisher, Gibco, Cat no. 16000044), 100 units penicillin and 100 µg ml$^{-1}$ streptomycin (Thermo Fisher, Gibco, Cat no. 16000044) and 1% NEAA (Thermo Fisher, Gibco, Cat no. 11140035) in a humidified $CO_2$ incubator at 5% $CO_2$ and 37 °C. HLF-1 cells were donated by Dr. Sara Rolandsson (Lund University). Human lung adenocarcinoma cells (A549) were maintained in F-12K medium (Thermo Fisher, Gibco, Cat no. 21127022) supplemented with 10% FBS (Thermo Fisher, Gibco, Cat no. 16000044), 100 units penicillin and 100 µg ml$^{-1}$streptomycin (Thermo Fisher, Gibco, Cat no. 16000044) in a humidified $CO_2$ incubator at 5% $CO_2$ and 37 °C. A549 cells were donated by Dr. Darcy Wagner (Lund University).

### MTT assay

The MTT (Formazan solution [MTT, Merck, Cat no. M2128]) cell viability assay was conducted to determine in vitro toxicity of A5, ETE-S, and ETE-PC (synthesized according to Gerasimov et al.[16]). Briefly, HLF-1 ($2 \times 10^4$ cells well$^{-1}$) were plated in a 96-well flat-bottom microplate and grown for 24 h. Cells were treated with A5, ETE-S, or ETE-PC (0–1000 µg ml$^{-1}$) for 24 h. Post-treatment, 200 µL of MTT (0.5 mg mL$^{-1}$) was added into each well and incubated at 37 °C for 4 h. After incubation, 200 µL of

isopropanol (Sigma-Aldrich 34863) was added to dissolve formazan crystals. The optical density of the formazan solution, as a measure of live cells, was obtained using a microplate reader at 570 nm (Spark Cyto, Tecan). The formazan signal (live cell count) was normalized to the control samples not exposed to A5 or ETE-Rs. 3 biological and 3 technical replicates were used for each setting. Each well contained up to 200 µg of our compound (1 mg mL$^{-1}$, 200 µL) which can be compared to a typical in vivo injection of 400 ng (40 mg mL$^{-1}$, 10 nL).

### In vivo−tail fin assay

Before microinjection, fish were anesthetized with tricaine medium (final concentration 0.2 mg/mL) until opercular movements had ceased and the fish did not respond to vibrations caused by tapping close to the tricaine container. An anesthetized fish was placed on its side on a plate filled with 1% agarose (Agarose, LE, Analytical Grade, Promega Corporation) in E3 medium that had been allowed to solidify. A piece of moist tissue paper was placed over the fish to keep the body from drying but still exposing the caudal fin. The plate was then transferred to the microinjection setup, and a glass capillary with a 30 µm diameter bevelled tip (cat. No. BM100T-15. Bevelled, straight, shortened + firepolished ends from Biomedical-Instruments GMBH) filled with polymer solution was used to inject the inter-rays of the caudal fin. The total injection volume per inter-ray was estimated to be in the 100 nL range. After injection, the fish were revived directly by flushing the gills with fresh aquarium water and transferred to a post-op aquarium for observation.

### In vivo−brain. Surgery and microinjection

Before surgery and microinjection, fish were anesthetized with tricaine medium (final concentration 0.2 mg/mL) until opercular movements had ceased and the fish did not respond to tail pinching. For surgery, the anesthetized fish was placed in a mold made of moist tissue paper for stabilization. Then, a small hole was made in the parietal bone just above the corpus cerebelli and immediately left of the midline with the tip of a 30 G needle. The fish was then transferred, in its tissue paper mold, to the microinjection setup, and a capillary filled with polymer solution (see below) with a 30 µm diameter bevelled tip (cat. No. BM100T-15. Bevelled, straight, shortened + firepolished ends from Biomedical-Instruments GMBH) was inserted through the hole in the skull roof to a depth of 700 µm. Then, three injections were made: one each at 700 µm, 500 µm, and 300 µm depths. The total injection volume was estimated to be 10 nL. After injection, fish were either revived directly by flushing the gills with fresh aquarium water and transferred to a post-op aquarium for observation or subjected to electropolymerization. We used adult (12–24 months old) zebrafish from the following strains: Casper mutant (Tg(elav3:GCaMP6f)) on nacre background, DAT Tg(dat:EGFP) on wild type AB background, and AB wild type.

### In vivo−brain. Electrofunctionalization

When the polymer injection was followed by electropolymerization, a counter electrode was placed on the skin at one of the nostrils, and an iridium-coated polymer-containing capillary served as the electrode. Injections were performed as described above; after the injection, the capillary was left in place in the brain. After 1 min−allowing the polymer solution to diffuse−the injected polymer was electropolymerized by applying 1.5 V vs Au counter electrode (0.5 V vs. Ag/AgCl counter electrode), current around 1–3 µA, over the electrodes for 5 min using a Keithley sourcemeter 2612B (Keithley Instruments). One side of the 30 µm diameter glass injection capillary (cat. No. BM100T-15. Bevelled, straight, shortened + firepolished ends from Biomedical-Instruments GMBH) was pre-coated with 50 nm Ir in a Quorum sputterer (QT 150, Quorum technologies) resulting in a conductive capillary with maintained backside optical access to verify liquid levels before injection.

When experiments were of longer out-of-water duration than 10 min, the fish was intubated for superfusion of the gill chambers with

aerated aquarium water containing 0.1 mg mL$^{-1}$ tricaine (ethyl 3-aminobenzoate methanesulfonate), using a Peri-Star Pro peristaltic pump (World Precision Instruments). Initially, we experimented with diffusion and electropolymerization durations longer than 5 min each. This did improve polymer spread and functionalization but did also affect the fish more negatively.

## Polymer formulations for microinjections
The following polymer formulations were used for microinjections into the brain (all dissolved in Millipore water): A5 (20 mg mL$^{-1}$); A5 (20 mg mL$^{-1}$) + ETE-S (40 mg mL$^{-1}$); A5 (20 mg mL$^{-1}$) + ETE-PC (40 mg mL$^{-1}$).

## Post-experiment tissue processing
After polymer injection, with or without electropolymerization, the fish was allowed to recover as described above and then transferred to aquaria for different post-injection survival times. Fish were sacrificed for histological examination or conductivity measurements after 1 h, 2 h, 3 h, 4 h, 1 day, 2 days, 3 days, 7 days, 8 days, 9 days, or 11 days. The fish were euthanized by immersion in ice-cold water for 10 min and then decapitated. The brains were either excised directly without fixation for freezing on dry ice in TissueTek OCT (Fisher scientific: epredia Neg-50) or processed immediately for redox staining (see below), or the skull roof was opened and the head (jaws removed) fixed overnight in 4% paraformaldehyde in 0.1 M phosphate buffer (Thermo Fisher J61899.AP).

Fresh frozen brains were cryosectioned in the sagittal plane (20–50 μm section thickness, depending on subsequent processing) in a Cryostar NX70 cryostat. Sections were mounted on Superfrost Gold microscope slides for microscopy, or on interdigitated gold electrodes for conductivity measurements.

Paraformaldehyde-fixed brains were excised from the skull, rinsed in phosphate-buffered saline (PBS), cryoprotected in PBS with 25% (w/v) sucrose, and frozen on dry ice in TissueTek OCT. The brains were cryosectioned in the sagittal plane (16 μm or 30–50 μm section thickness, dependent on subsequent processing). Sections were mounted on Superfrost Gold slides for further processing.

## Redox staining
For redox staining (protocol modified from Horowitz et al.[15]), the excised brains were placed in a solution containing 4% paraformaldehyde, 1 mM N-ethyl maleimide (Sigma-Aldrich E3876), 2 μM Alexa647-maleimide (Thermo Fischer A20347) and 0.05% Triton X-100 (Sigma-Aldrich X-100) in PBS for incubation overnight (ca. 16 h) at 8 °C. Then, the brains were rinsed in PBS (4 × 30 min), incubated in 5 mM TCEP (Sigma-Aldrich 75259) in PBS for 6 h, rinsed in PBS (5 × 15 min), and placed in a solution containing 1 mM N-ethyl maleimide and 2 μM Alexa555-maleimide (Thermo Fischer A20346) in PBS for incubation overnight (ca. 16 h) at 8 °C. Then, the brains were rinsed in PBS, cryoprotected in PBS with 25% (w/v) sucrose, and frozen on dry ice in TissueTek OCT (Fisher Scientific: epredia Neg-50). The brains were cryosectioned in the sagittal plane (30 μm section thickness). Sections were mounted on Superfrost Gold slides and coverslipped with Pro-Long Gold antifade reagent with DAPI (Thermo Fischer P36931) as the mounting medium. In the presented figures, fluorescence microscopy of the sections shows the extent of oxidized thiols (magenta), indicating tissue damage against a background of reduced thiols (cyan).

## Histology
For histological staining, we used 30 μm cryosections of paraformaldehyde-fixed and cryoprotected brains (see above). Slides with cryosections were air dried for at least 2 h at room temperature before histological staining. We used two standard staining protocols: Cresyl Violet (Nissl) stain (Sigma-Aldrich C5042) and Methylene Blue (Sigma-Aldrich 319112)–Thionin (Nissl) (Sigma-Aldrich 861340) stain. After staining, the slides were dehydrated in ethanol, cleared in xylene

(Merck 1.08661.1000), and coverslipped using Pertex as the mounting medium.

## NeuroTrace 640/660 staining
For selective fluorescent staining of neuronal cell bodies, slides with 30 μm thick cryosections were incubated with the fluorescent Nissl stain NeuroTrace 640/660 (Thermo Fisher N21483) diluted 1:50 in PBS for 30 min. After rinsing in PBS, the slides were coverslipped with ProLong Gold antifade reagent with DAPI (Thermo Fischer P36931) as the mounting medium.

## Immunofluorescence
For immunofluorescence, the fixed and cryoprotected brains (11 days post-injection survival; $n = 8$) were serially cryosectioned in the sagittal plane (16 μm section thickness). The serial cryosections were collected as two series (a and b) of alternate sections. Both series were processed for immunofluorescence. Series a was incubated 16 h at 4 °C with a neuronal marker, anti-microtubule associated protein 2 (MAP 2 (2a + 2b); mouse monoclonal antibody clone AP-20, Sigma-Aldrich) diluted 1:200. Series b was incubated 16 h at 4 °C with a radial glia marker, anti-glial fibrillary acidic protein (GFAP; mouse monoclonal antibody [ZRF-1] ab154474, Abcam) diluted 1:1200. After buffer rinses series a was incubated 1 h at room temperature with goat-anti-mouse IgG-Alexa Fluor 488 (Invitrogen A11001) diluted 1:200, and series b with goat-anti-mouse IgG-Alexa Fluor 488 (Invitrogen A11001) diluted 1:200 and the neuronal marker Neuro-Trace 640/660 fluorescent Nissl stain diluted 1:100. After buffer rinses the slides were coverslipped with ProLong Antifade Gold with DAPI (Thermo Fischer P36931) as mounting medium.

## Imaging
Freshly made cryosections were inspected without coverslipping for the presence of polymer and ETE-S monomer fluorescence using an Olympus CKX 41 microscope with 10x/0.40 and 20x/0.75 UPlanSApo, and 40x/0.85 UPlanApo objectives. For better resolution of the polymer structure in the unmounted sections, we used a 40x/0.80 W water immersion objective, with a drop of water directly on the brain section as an immersion medium. Images were obtained with a DMK 33UX174 monochrome camera (The Imaging Source). For multifluorescence imaging, we used either an Olympus IX73 fluorescence microscope with 10x/0.40 and 20x/.75 UPlanSApo objectives, and a Hamamatsu Orca R2 camera or a Nikon Ti2 fluorescence microscope with 40x/0,95 Plan Apochromate objective and a Nikon DS-Qi2 camera. For inspection and documentation of histologically stained sections, we used an Olympus CX63 microscope with E3CMOS (PIN:EP106300A) camera.

## Electrical measurements
Brain sections with polymer were placed on interdigitated Au electrodes connected to a Keithley sourcemeter 2612B (Keithley Instruments). Two of the interdigitated electrodes were contacted using external microelectrodes. An applied voltage was swept, and the resulting current was registered. This was repeated for all interdigitated electrode leads covering the conductive polymer. The distance between adjacent electrodes was 15 μm, and the width was 2.5 mm. Additional in vitro measurements of the ETE-Rs in organic electrochemical transistor set-ups can be found in the main article ref. 16 (Gerasimov et al.).

## Injection and brain slice preparation
A mixture of A5 and ETE-S was microinjected and electropolymerized in the brain, as described above, in adult zebrafish Casper mutants (Tg(elav3:GCaMP6f)). For these experiments, the capillary was cut immediately above the skull roof and left with the tip superficially in the brain. One day post-injection, the fish were euthanized by immersion in ice-cold aquarium water and decapitated. The brains were rapidly dissected and embedded in 3% low melting point agarose dissolved in Zebrafish normal Ringer solution. The blocks were cooled on a metal

plate and transferred to NMDG cutting solution on ice, trimmed, and mounted for vibratome sectioning. 300 and 400 μm sagittal sections containing the capillary tip and the polymer electrode in the tissue were cut from each brain, transferred to HEPES recovery solution, and allowed to reach room temperature (ca 24 °C). See Asrican and Song for detailed protocol[23]. The vibratome section was then transferred to artificial (zebrafish) cerebrospinal fluid (aCSF)[24], inspected for GFP positivity, and prepared for electrical stimulation and Ca$^{2+}$-imaging.

## Ca$^{2+}$ imaging in brain slices

Contacted brain slices were imaged in aCSF imaging medium using a 4x/0.10 Nikon objective with 30 mm working distance to allow simultaneous imaging and electrical stimulation. A 475 nm LED was used for excitation during the fluorescent imaging, and a DFK 33UX264 camera connected to the IC Measure software was used to capture the images.

## Electrical stimulation in brain slices

10 μm tungsten microelectrodes (Signatone, Gilroy CA) were allowed to contact the A5 in the brain slices, either directly or by contacting the injection capillary which in turn was contacted to the A5. A Grass S48 stimulator (Astro Med) was used to supply the stimulating square voltage pulses with the following settings: 2 trains per second, 200 ms train duration, 20 pulses per second, and 2 ms pulse duration. The magnitude of the voltage pulses as dialed in on the Grass stimulator ranged between 6–14 V. Large losses, including contact resistances, stray currents in buffers surrounding the tissue slices, poor impedance matching, and others motivate the need for a reasonably high stimulating voltage. We did not observe extensive bubble formation around the electrodes, which would be the case at high input powers.

## Preparation of vibratome slices for labeling and iDisco optical clearing

After Ca$^{2+}$ imaging, the vibratome slices were fixed in 4% paraformaldehyde in 0.1 M phosphate buffer overnight at 4 °C. After 2 × 30 min rinses in phosphate-buffered saline (PBS), the slices were permeabilized in PBS + 0.25% TritonX-100 (Sigma-Aldrich X-100) + 20% DMSO (1 h), PBS + 0.1% TritonX-100 + 20% DMSO (1 h), and PBS + 0.1% TritonX-100 (2 × 30 min). The slices were then incubated with the fluorescent Nissl stain NeuroTrace 640/660 (Thermo Fisher N21483) (2%) dissolved in PBS + 0.1% TritonX-100, for 3 days in the dark at 4 °C.

The slices were then rinsed in PBS (4 × 30 min) and embedded in 1% agarose blocks (for mounting in the light sheet microscope). After an additional rinse in PBS the samples were dehydrated in a graded methanol (Sigma-Aldrich 34860) series (20%, 40%, 60%, 80% methanol (in MQW; 30 min in each step), transferred to 100% methanol (2 × 30 min), and stored in 100% methanol overnight. Then, the samples were infiltrated with 66% dichloromethane (Sigma-Aldrich 270997) + 33% methanol for 3 h, rinsed in 100% dichloromethane (2 × 15 min), and transferred to 100% benzylether (DBE) (Sigma-Aldrich 108014) (for oxygen-free storage in the dark at 4 °C until microscopy.

## 3D imaging

Samples were imaged in a chamber filled with DBE. The cleared brain slice embedded in agarose was imaged on an Ultra Microscope II (LaVision Biotec) equipped with an sCMOS camera (Andor Neo, model 5.5-CL3) and 4x objective lenses (LaVision LVMI-Fluor 4x/0.3). We used two laser configurations (488 nm and 640 nm) with the following emission filters: 525/50 for endogenous background (blood vessels) and visualization of ETEs monomer and 680/30 nm for visualization of neurons (Neurotrace 640/660) (Thermo Fisher N21483). Stacks were acquired with ImspectorPro64 (LaVision Biotec) using 3 μm z-steps to acquire the volume in 3D. This image stack was stitched to visualize the brain slice in 3D with Arivis Vision 4D 3.5.0 (Arivis AG). The rendered movie was compiled in Final Cut Pro 10.4.3 (Apple Inc.).

## Mechanical measurements

0.6% agarose molded in Ringer buffer was injected with 3 μl A5 (20 mg ml$^{-1}$) + ETE-PC (40 mg ml$^{-1}$) using a Hamilton syringe. The A5–ETE-PC was electropolymerized at 1.75 V for 20 min. Cross-sections were cut and placed in a Biomomentum Mach-1 mechanical tester. The test was performed in indentation mode, with a 0.5 mm diameter spherical indenter at a speed of 0.01 mm/s. Indenter depth: 0.15, 0.3, and 0.45 mm. The test profile consisted of the following steps:
- Contact (0.1gf)
- 10 min wait to recover from contact
- 3 stress-relaxations
- 3 sinusoidal tests at 0.1, 1, and 4 Hz, respectively

The data was analyzed following Babaei[25] and Wang[26].

The resulting moduli and relaxation parameters are displayed in Table 1.

For the sinusoid at 0.45 mm depth. Moduli are in gf/mm (gram force/mm). The signal at 4 Hz was unreliable and excluded. Table 2 below summarizes the results.

The agarose and A5–ETE-PC were indistinguishable at 0.1 Hz. At 1 Hz, the A5–ETE-PC had a lower modulus. Looking at the component of the Modulus, the agarose showed a higher elastic response, whereas the A5–ETE-PC may have become more viscous.

## Statistics and reproducibility

No statistical method was used to predetermine sample size, but care was taken to minimize the number of animals used. In general, we aimed to adhere to 3 biological replicates per experiment. In the electrical measurements, short-circuited contacts were excluded from analysis (electrically and optically verified shorts). The investigators were not blinded to allocation during experiments and outcome assessment since animals with implanted polymer electrodes were easily recognized by the appearance of the dark blue polymer.

## Data availability

All data supporting the findings of this study are available within the article and its supplementary files. Any additional requests for

**Table 1 | Results from the mechanical measurements of electropolymerized A5–ETE-PC in an agarose matrix**

|  | Shear modulus (kPa) | Poroelastic relaxation (s) | Viscoelastic relaxation (s) |
|---|---|---|---|
| Agarose | 1.27 ± 0.76 | 1931.65 ± 62.83 | 11.05 ± 1.82 |
| A5–ETE-PC | 0.57 ± 0.10 | 1258.30 ± 618.25 | 7.34 ± 4.90 |

**Table 2 | Dynamical mechanical indentation measurements**

|  | 0.1 Hz | | | | 1 Hz | | | |
|---|---|---|---|---|---|---|---|---|
|  | Phase | Modulus | Elastic (E') | Storage (E") | Phase | Modulus | Elastic (E') | Storage (E") |
| Agarose | 16.4 | 0.61 | 0.58 | 0.17 | 30.9 | 1.0 | 0.86 | 0.52 |
| A5–ETE-PC | 18.3 | 0.5 | 0.48 | 0.17 | 27.2 | 0.58 | 0.52 | 0.27 |

The phase is the shift in response relative to the excitation frequency. E' and E" = Modulus. The modulus is defined as E = E + iE.

information can be directed to, and will be fulfilled by, the lead contact. Source data are provided with this paper.

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

## Acknowledgements

This study was accomplished within the Lund University Strategic Research Areas MultiPark and NanoLund. We thank Dr. Xenofon Strakosas for assistance with interdigitated electrodes, Dr. Darcy Wagner for access to A549 cells, Dr. Sara Rolandsson Enes for HLF–1 cells, Dr. Bengt Mattsson for assistance with light sheet imaging, Dr. Yuriy Pomeshchik for access to fluorescence microscopes, and Dr. Cedric Dicko for assistance with mechanical testing. Equipment within Lund Nano Lab (LNL) and Lund University Bioimaging Centre (LBIC) was used to enable this research. R.O. acknowledges the financial support from Swedish Research Council (2018-05258, and 2018-06197), Swedish Foundation for Strategic Research (RMX18-0083), the Swedish Research Council 3 R, and MultiPark. M.H. acknowledges the financial support from The Swedish Research Council, the Crafoord Foundation, the Trygger Foundation, and the Royal Physiographic Society in Lund.

## Author contributions

R.O. conceptualized the fundamental idea and study design. The development and execution of the methods used in this study were carried out by M.H., P.E., F.E., and R.O. The monomer and polymer syntheses were designed and conducted by A.H.M., D.B., M.A.S., and R.O. The experiments involving zebrafish were conducted by M.H., K.H., P.E., and F.E. All the work related to cells was handled by A.S.Y. Electrical characterization was conducted by M.H. The acquisition of financial support for the project was accomplished by R.O. and M.H. The project was supervised by R.O. The original draft of this manuscript was written by M.H. and R.O. All authors contributed to the review and editing of the manuscript.

## Funding

## Competing interests

The authors declare no competing interests.
