## [Peer review file · Nature Communications]

REVIEWER COMMENTS

Reviewer #2 (Remarks to the Author):

The manuscript describes the *in vivo* assembly of a bioresorbable conductive polymer ('electrode') using the PEDOT-S derivative A5 in a zebrafish brain model. The research group already reported on the chemistry of A5 (ref 13), while this manuscript focuses on the *in vivo* application. The authors claim to offer a minimally invasive method to deliver 'electrodes' to the brain to treat non-chronic pathologies such as cancer. The concept is interesting and may present advantages with respect to alternative approaches for neural interfaces or reported 'injecting electrodes' (ref DOI 10.1002/adhm.201900892).

The authors are to be praised for their strong commitment to the 3R, and I agree that zebrafish is a good model for exploratory research. On the other hand, in the field of neural interfaces, the anatomy of the model is as equally important as the physiology. Therefore, I don't share the authors' opinion that 'methodologies and workflows presented here are general and not confined to zebrafish'. The manuscript does elegantly show how A5 assembles in the zebrafish brain but fails to quantitatively show both the 'success rate' and clearly describe the assembled 'electrode' size. Furthermore, the authors don't comment on how this technology could be translated in larger models and, in particular, how the distance between the biased and counter electrode affects the voltages and currents required for the electropolymerization, including their impact on interposed tissue. Can the 'micrometre-sized' injected electrodes be safely translated to treat a brain tumour of several centimetres? How many injections are likely to be required? How are these 'electrodes' reached to provide power and control? So far the authors have only shown neuromodulation in an *ex vivo* model (brain slices).

The literature supports the author's conclusion that most of the acute inflammatory process in neural interfaces is linked to the injury secondary to the implantation trauma. The authors show this well in their figures. However, the chosen model and the methodology fail to investigate the role of chronic inflammation, gliosis and foreign body reaction caused by the injected polymer. This requires longer time points, different immunohistological markers and likely larger animal models.

Besides that Fig 2 needs the scale bars, I have little to say about the presentation quality. The paper is very well written, and the figures are clear and concise.

In conclusion, this manuscript represents a step-wise development in injectable electronics. It offers an *in vivo* application of the previously described A5 polymer. While the results are well presented, they are primarily qualitative and unlikely to be translatable to larger animal models or humans. Nevertheless, as per the authors, the paper offers good exploratory research in the use of this polymer, which will now require more robust validation in more appropriate animal models.

Reviewer #3 (Remarks to the Author):

Upon reviewing the manuscript, it is my opinion that the authors have demonstrated a commendable level of effort in their preparation of the manuscript. While a similar paper has been reported recently in Science, the research presented in this manuscript is clear and well-organized, making it easy to follow the authors' methodology, results, and conclusions. Based on the quality of the work presented,

I recommend this manuscript for publication in Nature Communications.

Furthermore, the editor requested me to review the authors' response to a previous reviewer's comments during the manuscript's review process at Nature. Based on my assessment, the authors have successfully addressed the majority of the questions raised by that reviewer. However, there remains one unanswered query regarding "Line 141: how did you confirm that the transient electrodes left healthy tissues in the fin?". In my opinion, the authors have not adequately addressed this concern. Therefore, I recommend that the authors remove the term "healthy" from this paper, as several evaluations need to be conducted before making such a claim.

Point-by-point answers:

Reviewer #1, see, Reviewer #3.

Reviewer #2 (Remarks to the Author):

Q: The manuscript describes the *in vivo* assembly of a bioresorbable conductive polymer ('electrode') using the PEDOT-S derivative A5 in a zebrafish brain model. The research group already reported on the chemistry of A5 (ref 13), while this manuscript focuses on the *in vivo* application. The authors claim to offer a minimally invasive method to deliver 'electrodes' to the brain to treat non-chronic pathologies such as cancer. The concept is interesting and may present advantages with respect to alternative approaches for neural interfaces or reported 'injecting electrodes' (ref DOI 10.1002/adhm.201900892).

A: The reviewer has once again highlighted an alternative technology, dubbed "injectrode," which we infer to be the reviewer's reference point for the state-of-the-art in this field. As a result, we've conducted a thorough analysis and comparison of this technology with the one we proposed in our manuscript. We note that the authors of the cited paper have established a company, Neuronoff (www.neuronoff.com).

1. The referenced paper claims to inject electrodes *in vivo*, as the title implies; however, the electrodes used in the study were implanted in swine cadavers, and swine and rats through open surgery. There is no example of injecting the electrode into a living animal. The authors claim that open surgery was chosen for demonstration purposes, but a more convincing demonstration would have been to inject first and then surgically expose the tissue.
2. Our approach introduces a temporary bioresorbable electrode, while the metal/polymer electrode presented in the referenced paper necessitates surgical removal.
3. The needle used in their study has a diameter of 1.2 mm, which is 40 times larger than ours. Such a large needle has been linked to a severe risk of vascular rupture when used in sensitive tissue such as the brain (see, e.g., <https://doi.org/10.1101/2020.09.21.306498>).
4. In contrast to our nanoparticle solution, their method essentially extrudes a preformed polymer mixed with conductive materials. At their company, Neuronoff, they have moved away from this mixture and instead inject a preformed silver wire. This is a significant factor when considering initial fluidity; we can accommodate a 1.2 mm needle, but it remains uncertain whether they can use a 30 μ m needle.
5. Regarding brain mechanics, our technology is comparable to natural brain tissue mechanics, whereas theirs is 50-100 times stiffer.
6. There is a lack of data regarding toxicity in their study. They use a two-phase system comprised of an injectable preformed polymer mixed with conductive elements like silver flakes. Upon injection into the tissue, conductivity is provided by these silver flakes. However, it's known that silver compounds can release silver ions which are both cytotoxic and antimicrobial [DOI: 10.3390/ijms20020449]. This poses significant challenges for its application in any clinical or preclinical setting.

In our paper, we present conductive polymer systems that are devoid of potentially harmful elements. These systems can be used as a single-phase system (A5 only) or through electropolymerization/electrofunctionalization with ETE-Rs, allowing us to alter the properties and the function of the electrode.

Q: The authors are to be praised for their strong commitment to the 3R, and I agree that zebrafish is a good model for exploratory research. On the other hand, in the field of neural interfaces, the anatomy of the model is as equally important as the physiology. Therefore, I don't share the authors' opinion that 'methodologies and workflows presented here are general and not confined to zebrafish'. The manuscript does elegantly show how A5 assembles in the zebrafish brain but fails to quantitatively show both the 'success rate' and clearly describe the assembled 'electrode' size. Furthermore, the authors don't comment on how this technology could be translated in larger models and, in particular, how the distance between the biased and counter electrode affects the voltages and currents required for the electropolymerization, including their impact on interposed tissue. Can the 'micrometre-sized' injected electrodes be safely translated to treat a brain tumour of several centimetres? How many injections are likely to be required? How are these 'electrodes' reached to provide power and control?

A: We have clearly failed to convince Reviewer 2 of the generality of our methodologies. Indeed, we agree with Reviewer 2 that the gross anatomy of zebrafish is different from humans, but most brain areas are represented in zebrafish. Zebrafish models are used extensively in most areas of medical research nowadays, including brain diseases. In this study, the main focus is on the interaction with neurons and the ability to externally stimulate the electrode, bioresorption, and inflammatory reactions. For this exploratory research, we believe that the zebrafish is an excellent model. Here, it is important to stress that no animal is optimal and that rodents are not small humans.

The A5 polymer, upon injection into brain tissue, self-assembles into a conductive structure, relying solely on the inherent cations present in the tissue (refer to Mousa et al., Chem Mater for further details). The ionic concentrations in the brains of zebrafish, and other vertebrates, such as mice or humans, are similar, but we've demonstrated our ability to account for variations in ionic concentration by tailoring the injection formulation to match different tissues. In the case of zebrafish, we successfully adjusted for the caudal fin and the brain, thereby ensuring the technique's translatability. The subsequent electrofunctionalization process does not necessitate any species-specific molecules or conditions. This is a departure from previous approaches to induce the self-assembly of passive electrodes, which either necessitated the use of genetically modified animals (reference 6 in the main article) or relied on co-injected plant enzymes to form conductive structures (reference 8 in the main article).

Just like the 'injectrode' study recommended by the reviewer, we have now conducted supplementary experiments on larger *ex vivo* mouse brains. This allowed us to confirm the translatability of our method. Notably, we even recorded higher currents (indicating reduced resistance) in the mouse brain compared to the zebrafish brains when we analyzed the conductivity on our interdigitated electrodes. We have included this data in the supplementary material; please refer to Figure S6.

Related to the electrode distances: After microinjection, the contacted microcapillary is surrounded by A5+ETE-R solution. When adding the electric field, the ETE-R around the microcapillary will be electropolymerized and thus form a close interface. This will enable the A5 (+ETE-R) to act as an extension of the microcapillary with only minor resistive losses. The electrical potential at the A5-ETE-R electrode will be the same as on the microcapillary. The distance to the counter-electrode will, therefore, not be as critical for the electropolymerization, but it will be important for use cases where the electric field strength in between the polymer and counter electrode is of interest.

We have performed additional experiments where A5+ETE-S was injected into an agarose gel mimicking a brain followed by electropolymerization at varying electrode

distances (up to 55 mm, corresponding to close to 20 zebrafish brains). In short, we could not observe that the electrode distance influenced the formation of dendrites (neither temporally, geometrically, nor current needed for formation). We have added this data as supplemental Figure S11.

Related to success rate: On row 784, we mention that 2 out of 3 fish showed the polymer electrode after 7 days in the electropolymerized cohort. For the 1day experiments, we saw polymer in all fish. We have now tested >hundred fish the overall success rate is >90%.

In our research, we can install polymer electrodes with minimal invasion. The idea of treating solid tumors is compelling and is a direction we are likely to explore in future investigations. The reviewer is right in pointing out that treating larger, irregularly shaped tumors (> 3.5 cm) with electrotherapy, in general, presents a significant challenge. Current technology, such as the Nanoknife, employs multiple (2-6, with 6 being used for larger tumors) solid electrodes of about 2mm in diameter, which aren't optimal for brain tissue. At this point, it is purely speculative, given the lack of data; however, unlike solid and substrate-bound electrodes, the fluidity of our electrode could potentially allow for a continuous flow to distribute the liquid electrode within the tumor before and during treatment, as opposed to multiple injections. This is an intriguing area of study that we, or others, hope to address based on the work presented here.

Q: So far the authors have only shown neuromodulation in an *ex vivo* model (brain slices)

A: The reviewer accurately notes that the direct impact on neuronal signaling was demonstrated *ex vivo*. However, the study includes an *in vivo* procedure, thus, the injection and formation of the electrode and installation of the external electrode. Subsequently, the fish was euthanized, and its brain was extracted and sectioned, with the electrode in place, to allow for observing neuronal signaling via fluorescence.

Q: The literature supports the author's conclusion that most of the acute inflammatory process in neural interfaces is linked to the injury secondary to the implantation trauma. The authors show this well in their figures. However, the chosen model and the methodology fail to investigate the role of chronic inflammation, gliosis and foreign body reaction caused by the injected polymer. This requires longer time points, different immunohistological markers and likely larger animal models.

A: We thank the reviewer for acknowledging the suitability of our model to investigate the initial inflammation. The longest investigated time point (9 days) was chosen since it was short enough to observe a conductive injected polymer but long enough to ensure that the electrode had started to resorb. That is, the bioresorption process was ongoing without any observable onset of inflammatory processes. In the redox staining, we did not observe any inflammation nor occurrence of any foreign body reactions, which should show up as deviating from normal brain tissue.

We performed additional long-term experiments where the fish were left to swim for 11 days with implanted electrodes, followed by immunohistological analysis. In short, zebrafish brains (n=8) were injected with A5+ETE-S followed by electropolymerization. After 11 days, the fish were sacrificed, brains excised, fixed, cryoprotected, cryosectioned, and used for immunofluorescent staining. After 11 days, it was no longer possible to observe the injection track nor any polymer in the tissue. It had been fully bioresorbed. We have added representative images of the injected brain region as supplemental Figure S12.

We do agree that it would be interesting to investigate possible chronic alterations to brain

tissue in larger animals, but that would fall far outside of the present study. It is important to note that we do not expect inflammation to suddenly show up in the late stages of the bioresorption process. If so, these inflammatory signals should show up at the start of the bioresorption process as well.

Q: Besides that Fig 2 needs the scale bars, I have little to say about the presentation quality. The paper is very well written, and the figures are clear and concise.

A: We thank the reviewer for this notion. We have updated Fig 2.

Q: In conclusion, this manuscript represents a step-wise development in injectable electronics. It offers an *in vivo* application of the previously described A5 polymer. While the results are well presented, they are primarily qualitative and unlikely to be translatable to larger animal models or humans. Nevertheless, as per the authors, the paper offers good exploratory research in the use of this polymer, which will now require more robust validation in more appropriate animal models.

A: We must respectfully disagree with the reviewer; especially from a chemical standpoint, the difference is considerable; a solution that self-assembles *in vivo*, and is possible to stimulate externally, and is bioresorbed. We refer you to our previous comment about applying this method to larger animal models and our comparison with the paper suggested by Reviewer 2. From a materials science perspective, we don't foresee any obstacles to translating this platform technology to any other animal or specific tissues outside of the brain since it solely depends on endogenous cations for A5 self-assembly. This is a departure from previous approaches used to achieve the self-assembly of even passive electrodes, which required either genetically modified animals or co-injected plant enzymes. While the latter represents a step forward in translatability, no mammalian enzymes have been utilized to form the electrode to date.

Reviewer #3 (Remarks to the Author):

Q: Upon reviewing the manuscript, it is my opinion that the authors have demonstrated a commendable level of effort in their preparation of the manuscript. While a similar paper has been reported recently in Science, the research presented in this manuscript is clear and well-organized, making it easy to follow the authors' methodology, results, and conclusions. Based on the quality of the work presented, I recommend this manuscript for publication in Nature Communications.

Furthermore, the editor requested me to review the authors' response to a previous reviewer's comments during the manuscript's review process at Nature. Based on my assessment, the authors have successfully addressed the majority of the questions raised by that reviewer. However, there remains one unanswered query regarding "Line 141: how did you confirm that the transient electrodes left healthy tissues in the fin?". In my opinion, the authors have not adequately addressed this concern. Therefore, I recommend that the authors remove the term "healthy" from this paper, as several evaluations need to be conducted before making such a claim.

A: We have changed "healthy" to "no visible damage."

REVIEWERS' COMMENTS

Reviewer #2 (Remarks to the Author):

Q: The manuscript describes the in vivo assembly of a bioresorbable conductive polymer ('electrode') using the PEDOT-S derivative A5 in a zebrafish brain model. The research group already reported on the chemistry of A5 (ref 13), while this manuscript focuses on the in vivo application. The authors claim to offer a minimally invasive method to deliver 'electrodes' to the brain to treat non-chronic pathologies such as cancer. The concept is interesting and may present advantages with respect to alternative approaches for neural interfaces or reported 'injecting electrodes' (ref DOI 10.1002/adhm.201900892).

A: The reviewer has once again highlighted an alternative technology, dubbed "injectrode," which we infer to be the reviewer's reference point for the state-of-the-art in this field. As a result, we've conducted a thorough analysis and comparison of this technology with the one we proposed in our manuscript. We note that the authors of the cited paper have established a company, Neuronoff (www.neuronoff.com).

1. The referenced paper claims to inject electrodes in vivo, as the title implies; however, the electrodes used in the study were implanted in swine cadavers, and swine and rats through open surgery. There is no example of injecting the electrode into a living animal. The authors claim that open surgery was chosen for demonstration purposes, but a more convincing demonstration would have been to inject first and then surgically expose the tissue.
2. Our approach introduces a temporary bioresorbable electrode, while the metal/polymer electrode presented in the referenced paper necessitates surgical removal.
3. The needle used in their study has a diameter of 1.2 mm, which is 40 times larger than ours. Such a large needle has been linked to a severe risk of vascular rupture when used in sensitive tissue such as the brain (see, e.g., <https://doi.org/10.1101/2020.09.21.306498>).
4. In contrast to our nanoparticle solution, their method essentially extrudes a preformed polymer mixed with conductive materials. At their company, Neuronoff, they have moved away from this mixture and instead inject a preformed silver wire. This is a significant factor when considering initial fluidity; we can accommodate a 1.2 mm needle, but it remains uncertain whether they can use a 30 μ m needle.
5. Regarding brain mechanics, our technology is comparable to natural brain tissue mechanics, whereas theirs is 50-100 times stiffer.
6. There is a lack of data regarding toxicity in their study. They use a two-phase system comprised of an injectable preformed polymer mixed with conductive elements like silver flakes. Upon injection into the tissue, conductivity is provided by these silver flakes. However, it's known that silver compounds can release silver ions which are both cytotoxic and antimicrobial [DOI: 10.3390/ijms20020449]. This poses significant challenges for its application in any clinical or preclinical setting.

In our paper, we present conductive polymer systems that are devoid of potentially harmful elements. These systems can be used as a single-phase system (A5 only) or through electropolymerization/electrofunctionalization with ETE-Rs, allowing us to alter the properties and the function of the electrode.

R: The reviewer appreciates the author's thorough and enlightening comparison between their proposed technology and the 'injectrode' technology. As initially suggested, the reviewer continues to acknowledge potential advantages of the author's technology over the 'injectrode'. For the reader's benefit, it would be recommended that the authors succinctly incorporate some of the comparison points into their manuscript, whether in the introduction or discussion sections. While the innovative aspect of the chemistry is recognized, it has indeed been discussed in the authors' previous work, and the application methodology bears resemblance to existing technologies such as 'injectrode'. Therefore, in the reviewer's perspective, this work represents more of an incremental advancement rather than a significant leap in the field than the authors imply.

Furthermore, the reviewer would like to assure the authors that there are no affiliations with the 'injectrode' research group or any direct involvement with their paper.

Q: The authors are to be praised for their strong commitment to the 3R, and I agree that zebrafish is a good model for exploratory research. On the other hand, in the field of neural interfaces, the anatomy of the model is as equally important as the physiology. Therefore, I don't share the authors' opinion that 'methodologies and workflows presented here are general and not confined to zebrafish'. The manuscript does elegantly show how A5 assembles in the zebrafish brain but fails to quantitatively show both the 'success rate' and clearly describe the assembled 'electrode' size. Furthermore, the authors don't comment on how this technology could be translated in larger models and, in particular, how the distance between the biased and counter electrode affects the voltages and currents required for the electropolymerization, including their impact on interposed tissue. Can the 'micrometre-sized' injected electrodes be safely translated to treat a brain tumour of several centimetres? How many injections are likely to be required? How are these 'electrodes' reached to provide power and control?

A: We have clearly failed to convince Reviewer 2 of the generality of our methodologies. Indeed, we agree with Reviewer 2 that the gross anatomy of zebrafish is different from humans, but most brain areas are represented in zebrafish. Zebrafish models are used extensively in most areas of medical research nowadays, including brain diseases. In this study, the main focus is on the interaction with neurons and the ability to externally stimulate the electrode, bioresorption, and inflammatory reactions. For this exploratory research, we believe that the zebrafish is an excellent model. Here, it is important to stress that no animal is optimal and that rodents are not small humans.

The A5 polymer, upon injection into brain tissue, self-assembles into a conductive structure, relying solely on the inherent cations present in the tissue (refer to Mousa et al., Chem Mater for further details). The ionic concentrations in the brains of zebrafish, and other vertebrates, such as mice or humans, are similar, but we've demonstrated our ability to account for variations in ionic concentration by tailoring the injection formulation to match different tissues. In the case of zebrafish, we successfully adjusted for the caudal fin and the brain, thereby ensuring the technique's translatability. The subsequent electrofunctionalization process does not necessitate any species-specific molecules or conditions. This is a departure from previous approaches to induce the self-assembly of passive electrodes, which either necessitated the use of genetically modified animals (reference 6 in the main article) or relied on co-injected plant enzymes to form conductive structures (reference 8 in the main article).

Just like the 'injectrode' study recommended by the reviewer, we have now conducted supplementary experiments on larger ex vivo mouse brains. This allowed us to confirm the translatability of our method. Notably, we even recorded higher currents (indicating reduced resistance) in the mouse brain compared to the zebrafish brains when we analyzed the conductivity on our interdigitated electrodes. We have included this data in the supplementary material; please refer to Figure S6.

Related to the electrode distances: After microinjection, the contacted microcapillary is surrounded by A5+ETE-R solution. When adding the electric field, the ETE-R around the microcapillary will be electropolymerized and thus form a close interface. This will enable the A5 (+ETE-R) to act as an extension of the microcapillary with only minor resistive losses. The electrical potential at the A5-ETE-R electrode will be the same as on the microcapillary. The distance to the counter-electrode will, therefore, not be as critical for the electropolymerization, but it will be important for use cases where the electric field strength in between the polymer and counter electrode is of interest.

We have performed additional experiments where A5+ETE-S was injected into an agarose gel mimicking a brain followed by electropolymerization at varying electrode distances (up to 55 mm, corresponding to close to 20 zebrafish brains). In short, we could not observe that the electrode distance influenced the formation of dendrites (neither temporally, geometrically, nor current needed for formation). We have added this data as supplemental Figure S11.

Related to success rate: On row 784, we mention that 2 out of 3 fish showed the polymer electrode after 7 days in the electropolymerized cohort. For the 1day experiments, we saw polymer in all fish.

We have now tested >hundred fish the overall success rate is >90%.

In our research, we can install polymer electrodes with minimal invasion. The idea of treating solid tumors is compelling and is a direction we are likely to explore in future investigations. The reviewer is right in pointing out that treating larger, irregularly shaped tumors (> 3.5 cm) with electrotherapy, in general, presents a significant challenge. Current technology, such as the Nanoknife, employs multiple (2-6, with 6 being used for larger tumors) solid electrodes of about 2mm in diameter, which aren't optimal for brain tissue. At this point, it is purely speculative, given the lack of data; however, unlike solid and substrate-bound electrodes, the fluidity of our electrode could potentially allow for a continuous flow to distribute the liquid electrode within the tumor before and during treatment, as opposed to multiple injections. This is an intriguing area of study that we, or others, hope to address based on the work presented here.

R: The authors have certainly illustrated the capability of their conductive polymers to interface with neural tissues effectively in a zebrafish model. However, there is a significant leap between proving effectiveness in a small model organism and demonstrating translatable potential to larger organisms, particularly humans. To adequately support claims of translatability, it would be beneficial to conduct similar experiments in larger models, such as pigs, sheep, or non-human primates. While I appreciate the supplementary experiments presented by the authors, the claim regarding potential translation in larger brain remains not a 'reasonable' speculation (i.e. A reasonable speculation is that the procedures would be more straightforward with larger brains, e.g. rodents and primates, especially external connections.).

A crucial concern is the ability to reach larger tissue areas (several centimetres) in a minimally invasive manner, an assertion that cannot be entirely substantiated using the zebrafish model. Despite the noteworthy findings presented in this study, the limitations and speculative nature of certain claims warrant a reconsideration of the scope and framing of the manuscript.

Given these factors, it might be beneficial for the authors to revise the manuscript to more closely align with the data currently available, perhaps focusing on the novel aspects of the conductive polymers in the zebrafish model.

Q: So far the authors have only shown neuromodulation in an ex vivo model (brain slices)

A: The reviewer accurately notes that the direct impact on neuronal signaling was demonstrated ex vivo. However, the study includes an in vivo procedure, thus, the injection and formation of the electrode and installation of the external electrode. Subsequently, the fish was euthanized, and its brain was extracted and sectioned, with the electrode in place, to allow for observing neuronal signaling via fluorescence.

R: The authors' acknowledgment of conducting ex vivo analysis for the impact on neuronal signaling is appreciated. As such, the reviewer recommends the authors to further clarify this aspect in their manuscript.

Q: The literature supports the author's conclusion that most of the acute inflammatory process in neural interfaces is linked to the injury secondary to the implantation trauma. The authors show this well in their figures. However, the chosen model and the methodology fail to investigate the role of chronic inflammation, gliosis and foreign body reaction caused by the injected polymer. This requires longer time points, different immunohistological markers and likely larger animal models.

A: We thank the reviewer for acknowledging the suitability of our model to investigate the initial inflammation. The longest investigated time point (9 days) was chosen since it was short enough to observe a conductive injected polymer but long enough to ensure that the electrode had started to

resorb. That is, the bioresorption process was ongoing without any observable onset of inflammatory processes. In the redox staining, we did not observe any inflammation nor occurrence of any foreign body reactions, which should show up as deviating from normal brain tissue.

We performed additional long-term experiments where the fish were left to swim for 11 days with implanted electrodes, followed by immunohistological analysis. In short, zebrafish brains (n=8) were injected with A5+ETE-S followed by electropolymerization. After 11 days, the fish were sacrificed, brains excised, fixed, cryoprotected, cryosectioned, and used for immunofluorescent staining. After 11 days, it was no longer possible to observe the injection track nor any polymer in the tissue. It had been fully bioresorbed. We have added representative images of the injected brain region as supplemental Figure S12.

We do agree that it would be interesting to investigate possible chronic alterations to brain tissue in larger animals, but that would fall far outside of the present study. It is important to note that we do not expect inflammation to suddenly show up in the late stages of the bioresorption process. If so, these inflammatory signals should show up at the start of the bioresorption process as well.

R: The authors are thanked for their diligence in conducting additional experiments and their willingness to address this reviewer's concerns. Their effort to extend the time frame of their study and their comprehensive immunohistological analysis adds valuable information to the manuscript. While I concur with the authors that longer-term studies would stretch beyond the current scope of this work, it's a line of investigation worth considering for future research. This will provide more insights into the potential for chronic alterations in the tissue or delayed inflammation that might arise in a larger animal model or in longer-term applications.

Q: Besides that Fig 2 needs the scale bars, I have little to say about the presentation quality. The paper is very well written, and the figures are clear and concise.

A: We thank the reviewer for this notion. We have updated Fig 2.

Q: In conclusion, this manuscript represents a step-wise development in injectable electronics. It offers an in vivo application of the previously described A5 polymer. While the results are well presented, they are primarily qualitative and unlikely to be translatable to larger animal models or humans. Nevertheless, as per the authors, the paper offers good exploratory research in the use of this polymer, which will now require more robust validation in more appropriate animal models.

A: We must respectfully disagree with the reviewer; especially from a chemical standpoint, the difference is considerable; a solution that self-assembles in vivo, and is possible to stimulate externally, and is bioresorbed. We refer you to our previous comment about applying this method to larger animal models and our comparison with the paper suggested by Reviewer 2. From a materials science perspective, we don't foresee any obstacles to translating this platform technology to any other animal or specific tissues outside of the brain since it solely depends on endogenous cations for A5 self-assembly. This is a departure from previous approaches used to achieve the self-assembly of even passive electrodes, which required either genetically modified animals or co-injected plant enzymes. While the latter represents a step forward in translatability, no mammalian enzymes have been utilized to form the electrode to date.

R: While it's recognized that the chemical innovation in this work is significant, its prior description in previous publications lessens the novelty in the context of this manuscript. Also, the 'method' of injecting electrodes has been previously described. While the authors argue for the translatability of their methodology, the evidence presented within this study falls short of substantiating many of the claims made. Without this crucial validation, the paper appears more as an incremental advancement in the realm of injectable bioelectronics using a different compound. This perspective doesn't undermine the potential future impact of this research, but rather emphasizes the necessity for further work to be conducted before claims of broader applicability can be fully justified.

Point-by-point answers:

REVIEWERS' COMMENTS

Reviewer #2 (Remarks to the Author):

Q: The manuscript describes the in vivo assembly of a bioresorbable conductive polymer ('electrode') using the PEDOT-S derivative A5 in a zebrafish brain model. The research group already reported on the chemistry of A5 (ref 13), while this manuscript focuses on the in vivo application. The authors claim to offer a minimally invasive method to deliver 'electrodes' to the brain to treat non-chronic pathologies such as cancer. The concept is interesting and may present advantages with respect to alternative approaches for neural interfaces or reported 'injecting electrodes' (ref DOI 10.1002/adhm.201900892).

A: The reviewer has once again highlighted an alternative technology, dubbed "injectrode," which we infer to be the reviewer's reference point for the state-of-the-art in this field. As a result, we've conducted a thorough analysis and comparison of this technology with the one we proposed in our manuscript. We note that the authors of the cited paper have established a company, Neuronoff (www.neuronoff.com).

1. The referenced paper claims to inject electrodes in vivo, as the title implies; however, the electrodes used in the study were implanted in swine cadavers, and swine and rats through open surgery. There is no example of injecting the electrode into a living animal. The authors claim that open surgery was chosen for demonstration purposes, but a more convincing demonstration would have been to inject first and then surgically expose the tissue.
2. Our approach introduces a temporary bioresorbable electrode, while the metal/polymer electrode presented in the referenced paper necessitates surgical removal.
3. The needle used in their study has a diameter of 1.2 mm, which is 40 times larger than ours. Such a large needle has been linked to a severe risk of vascular rupture when used in sensitive tissue such as the brain (see, e.g., <https://doi.org/10.1101/2020.09.21.306498>).
4. In contrast to our nanoparticle solution, their method essentially extrudes a preformed polymer mixed with conductive materials. At their company, Neuronoff, they have moved away from this mixture and instead inject a preformed silver wire. This is a significant factor when considering initial fluidity; we can accommodate a 1.2 mm needle, but it remains uncertain whether they can use a 30 μ m needle.
5. Regarding brain mechanics, our technology is comparable to natural brain tissue mechanics, whereas theirs is 50-100 times stiffer.
6. There is a lack of data regarding toxicity in their study. They use a two-phase system comprised of an injectable preformed polymer mixed with conductive elements like silver flakes. Upon injection into the tissue, conductivity is provided by these silver flakes. However, it's known that silver compounds can release silver ions which are both cytotoxic and antimicrobial [DOI: 10.3390/ijms20020449]. This poses significant challenges for its application in any clinical or preclinical setting.

In our paper, we present conductive polymer systems that are devoid of potentially harmful elements. These systems can be used as a single-phase system (A5 only) or through electropolymerization/electrofunctionalization with ETE-Rs, allowing us to alter the properties and the function of the electrode.

R: The reviewer appreciates the author's thorough and enlightening comparison between their proposed technology and the 'injectrode' technology. As initially suggested, the reviewer continues to acknowledge potential advantages of the author's technology over the 'injectrode'. For the reader's benefit, it would be recommended that the authors succinctly incorporate some of the comparison points into their manuscript, whether in the introduction or discussion sections.

While the innovative aspect of the chemistry is recognized, it has indeed been discussed in the authors' previous work, and the application methodology bears resemblance to existing technologies such as 'injectrode'.

Therefore, in the reviewer's perspective, this work represents more of an incremental advancement rather than a significant leap in the field than the authors imply.

Furthermore, the reviewer would like to assure the authors that there are no affiliations with the 'injectrode' research group or any direct involvement with their paper.

A: We have incorporated comparisons with the closest state-of-the-art in the introduction and discussions. The reviewer is correct that one part of the chemistry was disclosed in a previous publication, the discovery of A5 and its chemical characteristics. Here in we disclose that the water-dispersed nanoparticles assemble into a conductive structure in situ in vivo without specific triggers. In addition, the combination with the ETE derivative adds a new component of functionality to the chemistry.

Q: The authors are to be praised for their strong commitment to the 3R, and I agree that zebrafish is a good model for exploratory research. On the other hand, in the field of neural interfaces, the anatomy of the model is as equally important as the physiology. Therefore, I don't share the authors' opinion that 'methodologies and

workflows presented here are general and not confined to zebrafish'. The manuscript does elegantly show how A5 assembles in the zebrafish brain but fails to quantitatively show both the 'success rate' and clearly describe the assembled 'electrode' size. Furthermore, the authors don't comment on how this technology could be translated in larger models and, in particular, how the distance between the biased and counter electrode affects the voltages and currents required for the electropolymerization, including their impact on interposed tissue. Can the 'micrometre-sized' injected electrodes be safely translated to treat a brain tumour of several centimetres? How many injections are likely to be required? How are these 'electrodes' reached to provide power and control?

A: We have clearly failed to convince Reviewer 2 of the generality of our methodologies. Indeed, we agree with Reviewer 2 that the gross anatomy of zebrafish is different from humans, but most brain areas are represented in zebrafish. Zebrafish models are used extensively in most areas of medical research nowadays, including brain diseases. In this study, the main focus is on the interaction with neurons and the ability to externally stimulate the electrode, bioresorbtion, and inflammatory reactions. For this exploratory research, we believe that the zebrafish is an excellent model. Here, it is important to stress that no animal is optimal and that rodents are not small humans.

The A5 polymer, upon injection into brain tissue, self-assembles into a conductive structure, relying solely on the inherent cations present in the tissue (refer to Mousa et al., Chem mater for further details). The ionic concentrations in the brains of zebrafish, and other vertebrates, such as mice or humans, are similar, but we've demonstrated our ability to account for variations in ionic concentration by tailoring the injection formulation to match different tissues. In the case of zebrafish, we successfully adjusted for the caudal fin and the brain, thereby ensuring the technique's translatability. The subsequent electrofunctionalization process does not necessitate any species-specific molecules or conditions. This is a departure from previous approaches to induce the self-assembly of passive electrodes, which either necessitated the use of genetically modified animals (reference 6 in the main article) or relied on co-injected plant enzymes to form conductive structures (reference 8 in the main article).

Just like the 'injectrode' study recommended by the reviewer, we have now conducted supplementary experiments on larger ex vivo mouse brains. This allowed us to confirm the translatability of our method. Notably, we even recorded higher currents (indicating reduced resistance) in the mouse brain compared to the zebrafish brains when we analyzed the conductivity on our interdigitated electrodes. We have included this data in the supplementary material; please refer to Figure S6.

Related to the electrode distances: After microinjection, the contacted microcapillary is surrounded by A5+ETE-R solution. When adding the electric field, the ETE-R around the microcapillary will be electropolymerized and thus form a close interface. This will enable the A5 (+ETE-R) to act as an extension of the microcapillary with only minor resistive losses. The electrical potential at the A5-ETE-R electrode will be the same as on the microcapillary. The distance to the counter-electrode will, therefore, not be as critical for the electropolymerization, but it will be important for use cases where the electric field strength in between the polymer and counter electrode is of interest.

We have performed additional experiments where A5+ETE-S was injected into an agarose gel mimicking a brain followed by electropolymerization at varying electrode distances (up to 55 mm, corresponding to close to 20 zebrafish brains). In short, we could not observe that the electrode distance influenced the formation of dendrites (neither temporally, geometrically, nor current needed for formation). We have added this data as supplemental Figure S11.

Related to success rate: On row 784, we mention that 2 out of 3 fish showed the polymer electrode after 7 days in the electropolymerized cohort. For the 1day experiments, we saw polymer in all fish. We have now tested >hundred fish the overall success rate is >90%.

In our research, we can install polymer electrodes with minimal invasion. The idea of treating solid tumors is compelling and is a direction we are likely to explore in future investigations. The reviewer is right in pointing out that treating larger, irregularly shaped tumors (> 3.5 cm) with electrotherapy, in general, presents a significant challenge. Current technology, such as the Nanoknife, employs multiple (2-6, with 6 being used for larger tumors) solid electrodes of about 2mm in diameter, which aren't optimal for brain tissue. At this point, it is purely speculative, given the lack of data; however, unlike solid and substrate-bound electrodes, the fluidity of our electrode could potentially allow for a continuous flow to distribute the liquid electrode within the tumor before and during treatment, as opposed to multiple injections. This is an intriguing area of study that we, or others, hope to address based on the work presented here.

R: The authors have certainly illustrated the capability of their conductive polymers to interface with neural tissues effectively in a zebrafish model. However, there is a significant leap between proving effectiveness in a small model organism and demonstrating translatable potential to larger organisms, particularly humans. To adequately support claims of translatability, it would be beneficial to conduct similar experiments in larger models, such as pigs, sheep, or non-human primates. While I appreciate the supplementary experiments presented by the authors, the claim regarding potential translation in larger brain remains not a 'reasonable' speculation (i.e. A reasonable speculation is that the procedures would be more straightforward with larger brains, e.g. rodents and primates, especially external connections.).

A crucial concern is the ability to reach larger tissue areas (several centimetres) in a minimally invasive manner, an assertion that cannot be entirely substantiated using the zebrafish model. Despite the noteworthy findings presented in this study, the limitations and speculative nature of certain claims warrant a reconsideration of the

scope and framing of the manuscript.

Given these factors, it might be beneficial for the authors to revise the manuscript to more closely align with the data currently available, perhaps focusing on the novel aspects of the conductive polymers in the zebrafish model.

A: We have clarified this in the manuscript and toned down our claims related to scalability (last paragraph of discussion section).

Q: So far the authors have only shown neuromodulation in an ex vivo model (brain slices)

A: The reviewer accurately notes that the direct impact on neuronal signaling was demonstrated ex vivo. However, the study includes an in vivo procedure, thus, the injection and formation of the electrode and installation of the external electrode. Subsequently, the fish was euthanized, and its brain was extracted and sectioned, with the electrode in place, to allow for observing neuronal signaling via fluorescence.

R: The authors' acknowledgment of conducting ex vivo analysis for the impact on neuronal signaling is appreciated. As such, the reviewer recommends the authors to further clarify this aspect in their manuscript.

A: We have clarified this in the manuscript by adding "in vivo" to the following sentence on p13 "After in vivo injecting the mixture of A5 and ETE-S".

Q: The literature supports the author's conclusion that most of the acute inflammatory process in neural interfaces is linked to the injury secondary to the implantation trauma. The authors show this well in their figures. However, the chosen model and the methodology fail to investigate the role of chronic inflammation, gliosis and foreign body reaction caused by the injected polymer. This requires longer time points, different immunohistological markers and likely larger animal models.

A: We thank the reviewer for acknowledging the suitability of our model to investigate the initial inflammation. The longest investigated time point (9 days) was chosen since it was short enough to observe a conductive injected polymer but long enough to ensure that the electrode had started to resorb. That is, the bioresorption process was ongoing without any observable onset of inflammatory processes. In the redox staining, we did not observe any inflammation nor occurrence of any foreign body reactions, which should show up as deviating from normal brain tissue.

We performed additional long-term experiments where the fish were left to swim for 11 days with implanted electrodes, followed by immunohistological analysis. In short, zebrafish brains (n=8) were injected with A5+ETE-S followed by electropolymerization. After 11 days, the fish were sacrificed, brains excised, fixed, cryoprotected, cryosectioned, and used for immunofluorescent staining. After 11 days, it was no longer possible to observe the injection track nor any polymer in the tissue. It had been fully bioresorbed. We have added representative images of the injected brain region as supplemental Figure S12.

We do agree that it would be interesting to investigate possible chronic alterations to brain tissue in larger animals, but that would fall far outside of the present study. It is important to note that we do not expect inflammation to suddenly show up in the late stages of the bioresorption process. If so, these inflammatory signals should show up at the start of the bioresorption process as well.

R: The authors are thanked for their diligence in conducting additional experiments and their willingness to address this reviewer's concerns. Their effort to extend the time frame of their study and their comprehensive immunohistological analysis adds valuable information to the manuscript. While I concur with the authors that longer-term studies would stretch beyond the current scope of this work, it's a line of investigation worth considering for future research. This will provide more insights into the potential for chronic alterations in the tissue or delayed inflammation that might arise in a larger animal model or in longer-term applications.

A: We agree with the reviewers' assessment.

Q: Besides that Fig 2 needs the scale bars, I have little to say about the presentation quality. The paper is very well written, and the figures are clear and concise.

A: We thank the reviewer for this notion. We have updated Fig 2.

Q: In conclusion, this manuscript represents a step-wise development in injectable electronics. It offers an in vivo application of the previously described A5 polymer. While the results are well presented, they are primarily qualitative and unlikely to be translatable to larger animal models or humans. Nevertheless, as per the authors, the paper offers good exploratory research in the use of this polymer, which will now require more robust validation in more appropriate animal models.

A: We must respectfully disagree with the reviewer; especially from a chemical standpoint, the difference is considerable; a solution that self-assembles in vivo, and is possible to stimulate externally, and is bioresorbed.

We refer you to our previous comment about applying this method to larger animal models and our comparison with the paper suggested by Reviewer 2. From a materials science perspective, we don't foresee any obstacles to translating this platform technology to any other animal or specific tissues outside of the brain since it solely depends on endogenous cations for A5 self-assembly. This is a departure from previous approaches used to achieve the self-assembly of even passive electrodes, which required either genetically modified animals or co-injected plant enzymes. While the latter represents a step forward in translatability, no mammalian enzymes have been utilized to form the electrode to date.

R: While it's recognized that the chemical innovation in this work is significant, its prior description in previous publications lessens the novelty in the context of this manuscript. Also, the 'method' of injecting electrodes has been previously described. While the authors argue for the translatability of their methodology, the evidence presented within this study falls short of substantiating many of the claims made. Without this crucial validation, the paper appears more as an incremental advancement in the realm of injectable bioelectronics using a different compound. This perspective doesn't undermine the potential future impact of this research, but rather emphasizes the necessity for further work to be conducted before claims of broader applicability can be fully justified.

A: We have clarified this in the manuscript and altered our claims in the last paragraph of the discussion section.